



# 12 years of continuous atmospheric O₂, CO₂ and APO data from Weybourne Atmospheric Observatory in the United Kingdom

Karina E. Adcock[1], Penelope A. Pickers[1], Andrew C. Manning[1], Grant L. Forster[1,2], Leigh S. Fleming[1,3], Thomas Barningham[4], Philip A. Wilson[1], Elena A. Kozlova[5], Marica Hewitt[1], Alex J. Etchells[6], and Andy J. Macdonald[1]

[1]Centre for Ocean and Atmospheric Sciences, School of Environmental Sciences, University of East Anglia, Norwich, United Kingdom
[2]National Centre for Atmospheric Science, University of East Anglia, Norwich, United Kingdom
[3]Now at: GNS Science, Gracefield, Lower Hutt, 5040, New Zealand
[4]British Antarctic Survey, Natural Environment Research Council, Cambridge, United Kingdom
[5]Faculty of Environment, Science and Economy, University of Exeter, Exeter, United Kingdom
[6]Research and Specialist Computing, University of East Anglia, Norwich, United Kingdom

*Correspondence to*: Karina E. Adcock (Karina.Adcock@uea.ac.uk)

**Abstract.** We present analyses of a 12-year time series of continuous atmospheric measurements of O₂ and CO₂ at the Weybourne Atmospheric Observatory in the United Kingdom. These measurements are combined into the term Atmospheric Potential Oxygen (APO), a tracer that is conservative with respect to terrestrial biosphere processes. The CO₂, O₂ and APO datasets discussed are hourly averages between May 2010 and December 2021. We include details of our measurement system and calibration procedures, and describe the main long-term and seasonal features of the time series. The 2-minute repeatability of the measurement system is approximately ±3 per meg for O₂ and approximately ±0.005 ppm for CO₂. The time series shows average long-term trends of 2.40 ppm yr⁻¹ (2.38 to 2.42) for CO₂, -24.0 per meg yr⁻¹ for O₂ (-24.3 to -23.8) and -11.4 per meg yr⁻¹ (-11.7 to -11.3) for APO, over the 12-year period. The average seasonal cycle peak-to-peak amplitudes are 16 ppm for CO₂, 134 per meg for O₂, and 68 per meg for APO. The diurnal cycles of CO₂ and O₂ vary considerably between seasons. The datasets are publicly available at https://doi.org/10.18160/Z0GF-MCWH (Adcock et al., 2023) and have many current and potential scientific applications in constraining carbon cycle processes, such as investigating air-sea exchange of CO₂ and O₂, and top-down quantification of fossil fuel CO₂.

## 1 Introduction

Carbon dioxide (CO₂) and oxygen (O₂) vary in the atmosphere due to a range of processes including terrestrial biosphere exchange, combustion (e.g. fossil fuels and wild fires) and air-sea gas exchange with the oceans. Terrestrial biosphere fluxes of O₂ and CO₂ are strongly anti-correlated in photosynthesis and respiration, with a stoichiometric exchange ratio of 1.05 – 1.10 moles of O₂ consumed per mole of CO₂ produced (or vice versa) (Severinghaus, 1995; Keeling and Manning, 2014). CO₂ and O₂ are also anti-correlated in fossil fuel combustion where O₂ is consumed and CO₂ is produced. The stoichiometric exchange ratio for fossil fuel combustion varies depending on the fossil fuel type. Generally, the ratios are assumed to be 1.17 mol mol⁻¹ for coal, 1.44 mol mol⁻¹ for oil and petroleum, and 1.95 mol mol⁻¹ for natural gas (Keeling 1988a; Steinbach



et al., 2011), leading to a globally weighted average of about 1.38 mol mol$^{-1}$ (Keeling and Manning, 2014). Cement
production also produces $CO_2$ but does not affect $O_2$ and globally cement emissions are a minor proportion of fossil fuel $CO_2$
emissions (Jones et al., 2021).

At the ocean surface, $CO_2$ and $O_2$ are constantly exchanged between the gaseous phase in the atmosphere and the dissolved,
aqueous phase in the ocean. However, unlike with the terrestrial biosphere and fossil fuel combustion, there is no fixed
stoichiometric coupling between $CO_2$ and $O_2$ in ocean-atmosphere fluxes due to their different seawater chemistry
properties. When $CO_2$ dissolves in seawater, it dissociates into more soluble carbonate and bicarbonate ions, but there is no
corresponding process for $O_2$. In addition, while marine biological and ocean upwelling processes both result in anti-
correlated variations in dissolved $O_2$ and $CO_2$, temperature-induced solubility changes result in correlated variations.
Furthermore, largely because of the $CO_2$ seawater chemistry, the air-sea equilibration time is much slower for $CO_2$ (~1 year)
than for $O_2$ (~1 month) (Broecker and Peng 1974; Keeling et al., 1993). Due to these varied relationships between $CO_2$ and
$O_2$, measuring atmospheric $O_2$ concurrently to $CO_2$ provides a wealth of additional information on carbon biogeochemical
cycles than can be achieved by measuring $CO_2$ alone.

Making high-precision measurements of $O_2$ mole fractions in the atmosphere is technically challenging as variations in $O_2$
are relatively very small compared to the atmospheric background (variations of the order of 10 ppm against a background of
~210,000 ppm). As such, atmospheric $O_2$ mole fractions are typically reported as changes in the ratio of $O_2$ to $N_2$, relative to
a standard with a known $O_2/N_2$ ratio. Since variations of $\delta(O_2/N_2)$ are small, the values are multiplied by $10^6$ and expressed
in 'per meg' units (Keeling and Shertz, 1992). $O_2$ and $N_2$ mole fractions are affected by changes in trace gases, such as $CO_2$,
since mole fractions are relative to the total amount and therefore changing the total number of molecules in the air will
make it appear as if the amount of $O_2$ and $N_2$ are changing even when they are not. Reporting $O_2$ as the $O_2/N_2$ ratio
circumvents this issue. Natural variations in $N_2$ are much smaller than $O_2$ (Manning and Keeling, 2006) and therefore any
changes in the $O_2/N_2$ ratio can be assumed to be due to a change in $O_2$. In practice most analytical techniques in use do not
measure the $O_2/N_2$ ratio directly and therefore when measuring $O_2$, $CO_2$ must also be measured concurrently, and a
correction applied to account for changes in $CO_2$. For simplicity, in this paper we refer to $O_2$ variations as $O_2$ mole fraction
changes rather than $\delta(O_2/N_2)$ ratio changes.


The first high-precision atmospheric $O_2$ measurements were made by Keeling (1988b) using an interferometric analyser.
Since then, several other independent analytical techniques have been developed to measure $O_2$ to ppm-level precision and
gas handling techniques have been refined to improve repeatability and compatibility of $O_2$ measurements (see Keeling and
Manning, 2014 for a review of the techniques and gas handling protocols). The high-precision atmospheric $O_2$ measurement
network has expanded globally, although many regions remain under sampled, and includes field stations, ship and aircraft
platforms, with measurements being made by several research groups. There are now approximately 30 stations around the





world making long-term, regular flask sample and/or continuous in situ measurements of atmospheric $O_2$ (Keeling and Manning, 2014 and references therein; van der Laan, et al., 2014; Morgan et al., 2019; Tohjima et al., 2019; Nguyen et al., 2022; Ishidoya et al., 2022; tinyurl.com/2rpczfy9).


Atmospheric Potential Oxygen (APO) is a calculated term that is helpful to further investigate some of the processes that affect both $O_2$ and $CO_2$, for example ocean processes (e.g. Ishidoya et al., 2022, Tohjima et al., 2015, Resplandy et al., 2019, Nevison et al., 2020, Pickers et al., 2017) and fossil fuel combustion (Pickers 2016; Chevallier et al., 2021; Pickers et al., 2022). APO is the sum of atmospheric $O_2$, and atmospheric $CO_2$ multiplied by the $O_2$:$CO_2$ molar exchange ratio of terrestrial

biosphere-atmosphere exchange (Stephens et al., 1998). APO is thus conservative with respect to terrestrial biosphere processes and is influenced by most fossil fuel emissions, due to their different exchange ratios, and by air-sea gas exchange of $O_2$ and $CO_2$. APO is calculated according to:

$$\text{APO} = O_2 + \frac{\alpha_L \times (CO_2 - 350)}{S_{O_2}} \tag{1}$$

where $\alpha_L$ is the global average terrestrial biosphere exchange ratio, 1.10 mol mol$^{-1}$ (Severinghaus, 1995), $S_{O_2}$ is the standard

atmospheric mole fraction of $O_2$, 0.2095 (Machta and Hughes, 1970) and 350 is an arbitrary reference value used to express APO on the Scripps Institution of Oceanography (SIO), U.S.A. APO scale. $CO_2$ is the measured $CO_2$ mole fraction, expressed in parts per million (ppm), and $O_2$ is the measured $O_2$ reported as $\delta(O_2/N_2)$ ratio changes. Both APO and $O_2$ are expressed in per meg units.

We present continuous in situ measurements of atmospheric $O_2$ and $CO_2$ mole fractions from Weybourne Atmospheric Observatory (WAO). The WAO is a field station located on the north Norfolk coast in a rural part of the United Kingdom (52.95°N, 1.12°E, Fig. 1). The station was established in 1992 by the University of East Anglia (UEA) and is a United Nations World Meteorological Organization Global Atmosphere Watch (WMO/GAW) 'Regional' station, a UK National Centre for Atmospheric Science Atmospheric Measurement and Observation Facility (NCAS/AMOF), and a European

Integrated Carbon Observation System (ICOS) 'Class II' station. The $O_2$ and $CO_2$ measurement system is housed in an air-conditioned concrete building. The sample inlets sit atop a 10 m mast, ~27 m above sea level and ~50 m inland from the North Sea coastline. In addition to continuous measurements of atmospheric $O_2$ and $CO_2$ other species routinely measured include CO, $CH_4$, $H_2$, $O_3$, $N_2O$, $SF_6$, $^{222}$Rn, NO, $NO_2$, PM2.5, PM10, $SO_2$, $NH_3$, $\delta^{13}$C-$CO_2$, $\delta^{18}$O-$CO_2$, and $\delta^{17}$O-$CO_2$ as well as basic meteorological parameters (www.weybourne.uea.ac.uk).


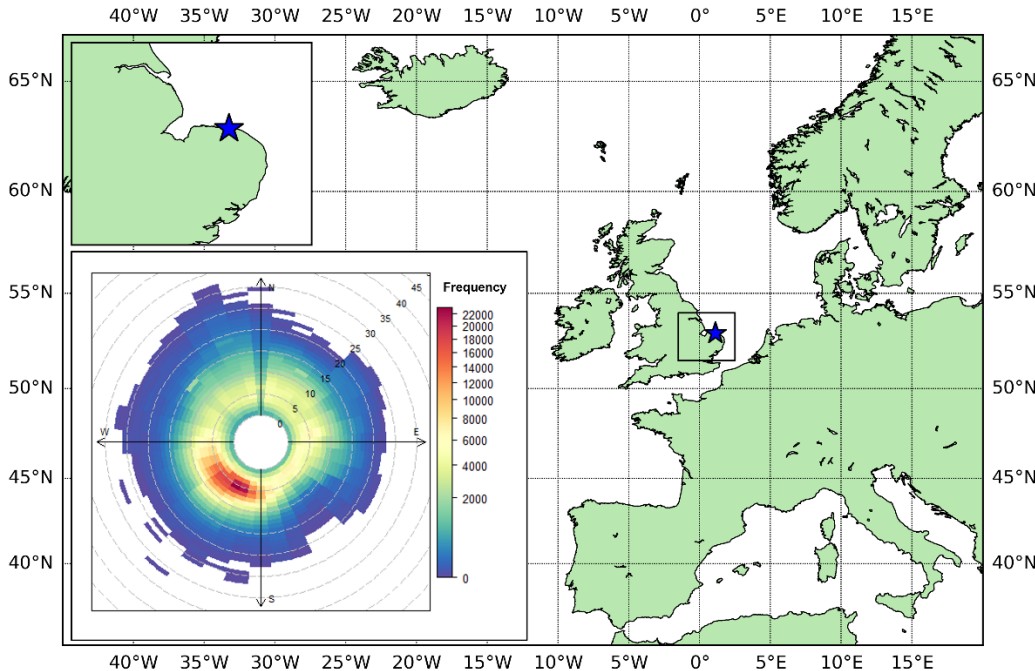

**Figure 1: Map showing the location of the Weybourne Atmospheric Observatory (WAO, 52.95°N, 1.12°E) as a blue star. Bottom left inset: Polar frequency plot showing wind speed (m s$^{-1}$) and wind direction at WAO averaged over the period 2016-2021. The frequency is the number of points with that wind speed and wind direction.**

In the remainder of this paper, Sect. 2 describes the measurement system and the seasonal decomposition methodologies, and Sect. 3 provides an assessment of the measurement system's performance via intercomparison programme results, and repeatability and compatibility measures. In Sect. 4, we present the main features of the dataset: long-term trends and interannual variability, and seasonal and diurnal cycles of $CO_2$, $O_2$ and APO. This analysis builds on work previously presented in the PhD theses of Wilson (2012) and Barningham (2018).

## 2 Methods

**Figure 2: Gas handling diagram of the Weybourne Atmospheric Observatory (WAO) $O_2$ and $CO_2$ measurement system. The drying, measurement, and calibration units are shown in separate boxes. The 'red' and 'blue' inlet lines are coloured accordingly, and the green colouring denotes the 'MKS' differential pressure gauge. The cylinders numbered 1-12 in the 'Blue Box' demonstrate the maximum capacity of the multi-position 'Valco' valve; these normally comprise of calibration cylinders, quality control cylinders referred to as Target Tanks (TTs), and cylinders that are periodically measured as part of intercomparison programmes. Also, in the Blue Box are cylinders referred to as Zero Tanks (ZTs) and Working Tanks (WTs); see main text for details.**

### 2.1 Analytical set up

The $CO_2$ and $O_2$ measurement system used at WAO is similar to those described in Stephens et al. (2007), Thompson et al. (2007) and Pickers et al. (2017). The WAO measurement system was first described in Patecki and Manning (2007) before it was deployed at WAO; it was also described in Pickers et al. (2022) and in the PhD theses of Wilson (2012) and
Barningham (2018). There have been many modifications and upgrades to the system since it was first installed. The following description and Fig. 2 present the system as it stands in 2023.

There are two inlet lines (Synflex, type 1300 tubing, ¼" OD) and the measurement system alternates sampling air between these two inlet lines every two hours to diagnose for leaks, blockages, or other faults. Each inlet line includes an aspirated air inlet, to avoid fractionation of $O_2$ molecules relative to $N_2$ (Blaine et al., 2006) and a small diaphragm pump (KNF Neuberger Inc.; model PM27653-N86ATE) to draw air through the inlet tubing.

Air from the inlets passes through a two-stage drying system to dry the sample air to <1 ppm water vapour, which prevents dilution effects caused by water vapour that would otherwise bias the $O_2$ measurements (Stephens et al., 2007). The first stage of the drying system is a Peltier element thermo-electric cooler (Tropicool, model XC3000A), set at approximately 1 °C. Water condenses out of the air and is drained away by peristaltic pumps (Cole Parmer, Masterflex). The Tropicool consists of stainless steel traps filled with 4 mm diameter Pyrex glass beads (to provide greater surface area to promote water
condensation). Traps are positioned both upstream and downstream of the diaphragm pump; the downstream pump provides better water condensation because of the above ambient pressure provided by the pump (~3 bar), whereas the upstream pump prevents water build up in the pump itself.  The second stage of the drying system is a VT490D cryogenic cooler (SP Scientific), which contains traps filled with 4mm diameter Pyrex glass beads in a 4 L ethanol bath, and achieves a dew point of approximately -80 °C.


The $CO_2$ mole fraction is measured by a non-dispersive infrared (NDIR) $CO_2$ analyser from Siemens Corp., model Ultramat 6E and the $O_2$ mole fraction is measured by an 'Oxzilla', a dual fuel cell $O_2$ analyser from Sable Systems International Inc. The fuel cells contain a gas-permeable membrane across which $O_2$ from the air permeates and undergoes electrochemical reduction in the cells, which contain a lead anode in an acidic electrolyte solution. The Ultramat and Oxzilla analysers are
both differential analysers and are placed in series. Sample air flows through one side of the analysers and air from the Working Tank (WT, see Sect. 2.2) flows continuously through the other side of the analysers; air from calibration, quality control and intercomparison cylinders passes through the 'sample' side. The pressures and flow rates are carefully balanced to be the same on both the sample air and WT air sides of the analysers. This balance is achieved with a differential pressure transducer (MKS Instruments, model Baratron 223B, ±10 mbar full scale range) and the two manual needles (Brooks
Instrument, model 8504) valves immediately downstream of the Ultramat (Fig. 2).

A solenoid valve (Numatics, model TM series) immediately upstream of the Oxzilla switches the sample air and WT air between each of the fuel cells every 60 seconds. Switching between sample air and WT air helps to eliminate short-term drift and increases the signal to noise ratio, since the amplitude of the fuel cell difference is doubled, but the noise remains the
same (Stephens et al., 2007). The first 30 seconds of data after every switch are discarded. Using the remaining 30 seconds

of data, the difference between the fuel cells is calculated. We refer to the fuel cell difference as the 'double differential $O_2$ value' and it is effectively twice the $O_2$ mole fraction (Stephens et al., 2007). The measurement system reports an $O_2$ and $CO_2$ measurement every 2 minutes.

As shown in Fig. 2, the measurement system also includes several pressure transducers, flow meters, solenoid valves, a flow controller, and temperature sensors (not shown in Fig. 2) that are all connected to an electronics control box (custom built in-house). Bespoke software (custom built in-house) enables the automation of many of the routine processes and minimizes human intervention. The software's automation functionality includes: measuring cylinders at predetermined intervals; switching between the two inlet sample lines; flushing sample or cylinder air prior to analysis, processing analyser signals to
calculate calibrated mole fractions in near real-time; rejecting calibrations outside a set range and notifying the user when key diagnostics stray outside of acceptable ranges. The software also records all of the analyser and diagnostic data (i.e. pressure, flow and temperature data) and routinely backs up all data files. In addition, the system can be accessed and controlled remotely, with the software displaying most of the data in near real-time.

**2.2 Calibration procedures**

The measurement system includes a calibration unit, consisting of a thermally insulated housing for gas cylinder stabilisation, a so-called 'Blue Box', where high-pressure cylinders are stored horizontally, a requirement for high-precision $O_2$ measurement (Keeling et al., 2007) that minimizes thermal and gravitational fractionation of $O_2$ and $N_2$ molecules inside the cylinders. Our calibration procedures are similar to those detailed for $O_2$ and $CO_2$ in Kozlova and Manning (2009).
Stored in the 'Blue Box' are Working Tanks (WTs), Working Secondary Standards (WSSes), Zero Tanks (ZTs) and Target Tanks (TTs), and occasionally cylinders that are part of intercomparison programmes.

A cylinder containing dry, compressed air, known as a Working Tank (WT), is continuously run through the reference side of the differential Ultramat and Oxzilla analysers. $CO_2$ and $O_2$ are measured by comparing the $CO_2$ and $O_2$ in the air from the
inlets (i.e. the sample side) to the $CO_2$ and $O_2$ in the air from the WT. By measuring the difference, any analyser response variability independent of the atmospheric mole fractions, for example from changes in room temperature or atmospheric pressure, is largely mitigated, since these changes will affect both the inlet air and WT air to roughly the same degree (Stephens et al., 2007).

A suite of three Working Secondary Standards (WSSes), which have high, medium and low mole fractions of $O_2$ and $CO_2$ spanning the typical range of mole fractions observed at the station, is used to calibrate the analysers every 47 hours (see calibration scale information below). The Ultramat exhibits a non-linear response to $CO_2$, so a quadratic equation ($y = ax^2 + bx + c$) is used in order to appropriately fit the analyser response function. The Oxzilla exhibits a linear response, so a linear



equation is used (y = bx + c). With the exception of the $CO_2$ c-term, the calibration coefficients are redetermined every 47
hours, and then these new values are used until the next calibration.

The $CO_2$ c-term (sometimes called the zero coefficient) is redetermined every 4 hours using a Zero Tank (ZT), which is a
cylinder filled with ambient air, and is run immediately after every calibration and then every 3 or 4 hours afterwards. This
$CO_2$ c-term adjustment is done to correct for baseline response drift in the Ultramat analyser, which is predominately due to
room temperature fluctuations in-between calibrations. The ZT correction is only required for the $CO_2$ measurements, as the
double differential switching accounts for analogous short-term temperature related drift in the $O_2$ analyser. A Target Tank
(TT), also filled with ambient air, is typically measured every 11 hours as a quality control check on both the measurement
system and the calibration procedures.

All cylinders used are 40L or 50L high-pressure aluminium cylinders (Luxfer Gas Cylinders Inc.) filled with very dry air (<1
ppm $H_2O$ content) to ~200 bar, at the Cylinder Filling Facility (CFF) at the UEA before being deployed at WAO. The $O_2$
and $CO_2$ mole fractions in the WSSes and the TT are pre-determined at UEA's Carbon Related Atmospheric Measurement
(CRAM) Laboratory, by measuring the cylinders using a bespoke Vacuum Ultraviolet (VUV) absorption analyser (Stephens
et al., 2003) for $O_2$ and a Siemens Ultramat 6F analyser for $CO_2$, against a suite of primary standards obtained from NOAA
(National Oceanic and Atmospheric Administration), U.S.A. and Scripps Institution of Oceanography (SIO) (see calibration
scale information below). The WSSes and TTs are used at WAO until the cylinder pressure is ~30 bar and the WTs and ZTs
are used until the cylinder pressure is ~5 bar, as there is frequently outgassing of gases from cylinder walls at low cylinder
pressures.

From its inception to June 2014, the WAO $CO_2$ data were reported on the SIO $CO_2$ calibration scale. From July 2014 to
December 2021 the $CO_2$ data were reported on the WMO $CO_2$ X2007 scale. Presently, all the data from October 2007 to
December 2021 have been transferred onto the WMO $CO_2$ X2019 scale using Equation 6 in Hall et al., (2021). The WMO
$CO_2$ X2019 calibration scale is maintained by the Central Calibration Laboratory (CCL) at the NOAA Earth System
Research Laboratories (ESRL) Global Monitoring Laboratory (GML). The $O_2$ measurements are reported on the SIO 'S2'
scale that was used by SIO from April 1995 to August 2017 (Keeling et al., 2020). For $O_2$ measurements there is no formally
recognised scale but most laboratories within the global $O_2$ community use or are linked to the SIO scale.

The measurement system was first installed at WAO in October 2007, however, between October 2007 and May 2010 the
system was still being developed and underwent major changes, leading to large gaps in the dataset and much of the data
being of unknown quality and with calibration scale issues, therefore the data from this period have been excluded and only
data from May 2010 to December 2021 are publicly available and are discussed in this paper.



## 2.3 Seasonal decomposition

The baselines of all three species were calculated from the hourly averages with the Robust Extraction of Baseline Signal (REBS) method using the RFBaseline function within the IDPMisc package in R (Ruckstuhl et al., 2012; 2020). The

RFBaseline function has three adjustable parameters: span, bi-weight function (b), and the number of iterations (maxit). The span is the smoothing time period, based on the frequency of data points. We have used a value of 0.01, which applies a smoothing window with a period of approximately four weeks, which we think is the most appropriate for distinguishing regional and background variability. The bi-weight function uses asymmetrical weighting, which is appropriate for $CO_2$ and $O_2$, where short-term excursions from the baseline are mostly uni-directional (positive for $CO_2$; negative for $O_2$), here we

used 0.01 and the number of iterations used is (4,0).

The time series were decomposed using the REBS results and STL (Seasonal Trend decomposition using LOESS (locally weighted scatterplot smoothing); (Cleveland et al., 1990)), so that salient features in long-term trends and seasonality can be described and quantified. The STL function (in R) performs a moving average calculation to separate the trend, seasonal and random components of the time series. STL has two smoothing parameters: s.window, which is the number of years to use

when estimating the seasonal component, and t.window, which is the number of consecutive observations to use when estimating the trend. We used an s.window of 5 years (this value is often used in other studies (Pickers and Manning, 2015)) and a t.window of 13149. Cleveland et al. (1990) suggests that the t.window should be approximately 1.5 to 2 times the number of observations in each year, for hourly data, that is 8766 hours per year (365.25 * 24 = 8766), and 1.5 * 8766 = 13149. The decomposed time series are shown in the supplement (Figs. S2-S4).


Since the currently available version of STL cannot handle gaps in the time series, the REBS results were interpolated using the na_seadec function in the imputeTS package in R (algorithm = 'interpolation', option = 'spline'), prior to decomposition using STL. Gap filling is to take account of times when sample air is not being measured because of routine cylinder analyses, because of scheduled system maintenance, or because of inadvertent downtimes owing to system faults. Figure S1

shows the baseline data with the interpolated values. In order to avoid possible end effects with the STL decomposition (Pickers and Manning, 2015), the time series were extended by a year both forwards and backwards in time, by taking the interpolated data for the first year and the last year and adding/subtracting the average annual trends for each species ($CO_2$: 2.31 ppm yr$^{-1}$, $O_2$: -24.1 per meg yr$^{-1}$, APO: -12.0 per meg yr$^{-1}$). These average annual trends were from an initial run of STL. After the seasonal decompositions were completed, these extended years were removed for all subsequent analyses

(including recomputing average annual trends). The baselines, the interpolation of the datasets and the seasonal decompositions presented here are just one example of the methods and parameters that could be used, since time series decomposition can be susceptible to the choice of method and parameters (Pickers and Manning, 2015).



## 3 Repeatability and compatibility of the measurement system

The WMO/GAW sets compatibility goals for measurements of greenhouse gas and related species in the atmosphere based
on what compatibility is scientifically desirable. Compatibility is a measure of the persistent bias between measurement
records (Crotwell et al., 2020). These WMO compatibility goals are ±0.1 ppm for $CO_2$ in the northern hemisphere and ±2 per
meg for $O_2$ globally. However, given current analytical capabilities, the $O_2$ goal is considered somewhat aspirational, as it is
not routinely achievable. The WMO also provide an extended compatibility goal of ±10 per meg for studies where greater
accuracy is not required (Crotwell et al., 2020), and which is routinely achievable for some laboratories. Nevertheless, for
most applications of $O_2$ data from our WAO station, the extended compatibility goal is not sufficient, so we strive to get as
close to the ±2 per meg goal as we can. We can also quantify repeatability from our datasets. Repeatability is a measure of
the closeness of the agreement between the results of successive measurements over a short period of time (Crotwell et al.,
2020). Repeatability goals should be one half the network compatibility goals, that is, ±0.05 ppm for $CO_2$, and ±1 per meg
for $O_2$ (or ±5 per meg for the extended $O_2$ goal). We have several ways in which to quantify compatibility and repeatability
from our WAO datasets, which we discuss in this section, including results from intercomparison programmes, analysing
data from our TT, ZT and WT measurements, and examining the standard deviation of sample air data.

### 3.1 Intercomparison programmes

Since 2010, WAO has participated in three intercomparison programmes in order to assess the compatibility of WAO with
other measurement sites. These intercomparison programmes are: 'Cucumbers' (www.cucumbers.uea.ac.uk); 'GOLLUM'
(Global Oxygen Laboratories Link Ultra-precise Measurements, www.gollum.uea.ac.uk); and the WMO/IAEA Round Robin
Comparison Experiment (World Meteorological Organization/International Atomic Energy Agency,
gml.noaa.gov/ccgg/wmorr/index.html). These intercomparison programmes involve measuring the same high pressure
cylinders filled with dry air, circulated amongst different laboratories and field stations, and comparing the measurements.
Each programme includes several trios of cylinders circulating amongst participants, except for the most recent WMO round
("Round 6"), where pairs of cylinders were circulated. The Cucumbers programme took place between 2005 and 2016 and
included nine atmospheric species, including $O_2$ and $CO_2$. The GOLLUM programme was initiated in 2004 to focus on $O_2$
compatibility, with participating laboratories and field stations measuring $O_2$ and $CO_2$. The WMO Round Robin programme
started in 1984 and primarily focuses on $CO_2$, but optionally includes other species, one of which is $O_2$. The intercomparison
results from WAO are shown in Fig. 3 and are plotted as the values measured at WAO minus the values from the first time
the cylinders were measured at the central laboratories: Max Planck Institute for Biogeochemistry (MPI-BGC) for the
Cucumbers; SIO for the GOLLUMs; and for the WMOs it is the NOAA/ESRL/GML for $CO_2$ and the National Center for
Atmospheric Research (NCAR), U.S.A. for $O_2$.





For $CO_2$, the average offsets at WAO relative to the initial central laboratory analysis are -0.04 ± 0.11 ppm for the

GOLLUMs, -0.11 ± 0.15 ppm for the Cucumbers and -0.22 ± 0.08 for the WMOs. These average offsets are the average of the cylinder differences and the ±1σ standard deviation of the cylinder differences. The average offset for the GOLLUMs is within the WMO compatibility goal (±0.1 ppm) but the Cucumbers and WMOs average offsets are larger. The GOLLUMs and Cucumbers ±1σ standard deviations are slightly larger than ±0.1 ppm, which implies that the WAO $CO_2$ measurements are not of sufficiently high precision to have confidence in the average offset. The WMOs ±1σ standard deviation is less than

±0.1 ppm, suggesting that the WAO $CO_2$ measurements are offset from the NOAA $CO_2$ values, although this is based on only two sets of measurements (in 2010 and 2015). For the Cucumbers, 16 out of the 30 sites involved in the programme, had a ±1σ standard deviation >±0.1 ppm (Manning et al., 2014). This demonstrates that, even for a gas such as $CO_2$, where measurement precision is routinely achievable to WMO requirements, maintaining such compatibility at field stations over the long-term is still challenging owing to other gas handling and calibration effects, such as leaks, drifting mole fractions

within cylinders, and disturbances to the ambient environment such as room temperature control issues (Manning et al., 2014).

For $O_2$, the average offsets at WAO relative to the initial central laboratory analysis are -0.5 ± 6.0 per meg for the GOLLUMs, 5.8 ± 12.3 per meg for the Cucumbers, and -2.6 ± 2.6 per meg for the WMOs. The GOLLUMs average offset is

within the ±2 per meg goal. The WMOs and Cucumbers average offsets are not within the ±2 per meg goal but are within the ±10 per meg extended goal. The WMO average offset is based on only one set of measurements of two cylinders ($O_2$ analysis was not an option with the WMO programme prior to Round 6). For Cucumbers none of the five sites measuring $O_2$ were within the ±2 per meg goal and four of the five sites were within the ±10 per meg goal (Manning et al., 2014). The GOLLUMs and WMOs ±1σ standard deviations are not within the ±2 per meg goal but are within the ±10 per meg goal. The

Cucumbers ±1σ standard deviation is not within the ±10 per meg goal. The measurements made in 2015 have larger ±1σ standard deviations, since during this time the WAO $O_2$ analyser was performing less well (see Sect. 3.2). The GOLLUM cylinders tend to perform better than the Cucumber cylinders for $O_2$, possibly because Cucumbers was not specifically focused on $O_2$ and therefore the laboratories that were not doing $O_2$ measurements may not have treated the cylinders in the way that is typically used for $O_2$ measurements (e.g., vertical storage and non-ideal pressure regulators) and that this could

have influenced the $O_2$ mole fraction in the cylinders. Some of the intercomparison cylinders had $O_2$ and $CO_2$ mole fractions outside of the calibration range of our measurement system, which may have influenced the results. While in general the intercomparisons show some variation, there does not appear to be any systematic drift over time, indicating long-term stability of the WAO $O_2$ and $CO_2$ calibration scales. The results of the intercomparison programmes should be kept in mind when comparing WAO data to other stations.







**Figure 3: Cucumbers, GOLLUM and WMO intercomparison cylinder results for $CO_2$ and $O_2$ plotted as the WAO measurements minus the initial central laboratory measurements. The horizontal shaded grey bands represent the WMO compatibility goals: ±0.1 ppm for $CO_2$, and ±2 per meg and ±10 per meg for $O_2$. The legends state the unique cylinder ID numbers. If the same cylinder was measured more than once throughout the programme, this is indicated**
**by connecting symbols with lines. The error bars show the ±1σ standard deviation of the cylinder measurements and in some cases the error bars are smaller than the symbols. All programmes involve trios of cylinders, with the exception of 'Round 6' of the WMO programme for which there were pairs of cylinders. WAO was originally part of the 'Inter-1' Cucumbers rotation but later moved to the 'Euro-6' rotation; WAO participated in both the 'Frodo' and 'Bilbo' GOLLUM rotations concurrently; and WAO participated in 'Round 5' and 'Round 6' of the WMO**
**Round Robins. The x-axis tick marks are at the beginning of each year.**

### 3.2 Stability of Target Tank mole fractions

The Target Tank (TT) is used to check the performance of the $O_2$ and $CO_2$ measurement system and is typically measured every 11 hours. The TT has a dual purpose: firstly, it is used to check how compatible the measurements at WAO are,
compared to the UEA CRAM Laboratory, where TT cylinders are usually analysed prior to being put into use at WAO. Secondly, the TT can also indicate the repeatability of the WAO $CO_2$ and $O_2$ analysers. Between May 2010 and December 2021, eleven TTs were measured at WAO, with a total of 8365 TT measurements, where we define a TT measurement as the mean of 7 consecutive two-minute measurements. TT measurements made when there were known technical issues have been removed, resulting in 961 $O_2$ and 878 $CO_2$ TT measurements removed, leaving 89% of the $O_2$ data and 90% of the $CO_2$
data. The TT does not pass through the inlet lines or the first stage of the drying system (Fig. 2) and therefore the repeatability of the TT measurements is not a quality control check on the whole measurement system but of the analysers, the internal calibration, and the gas handling system from the TT to the analysers.

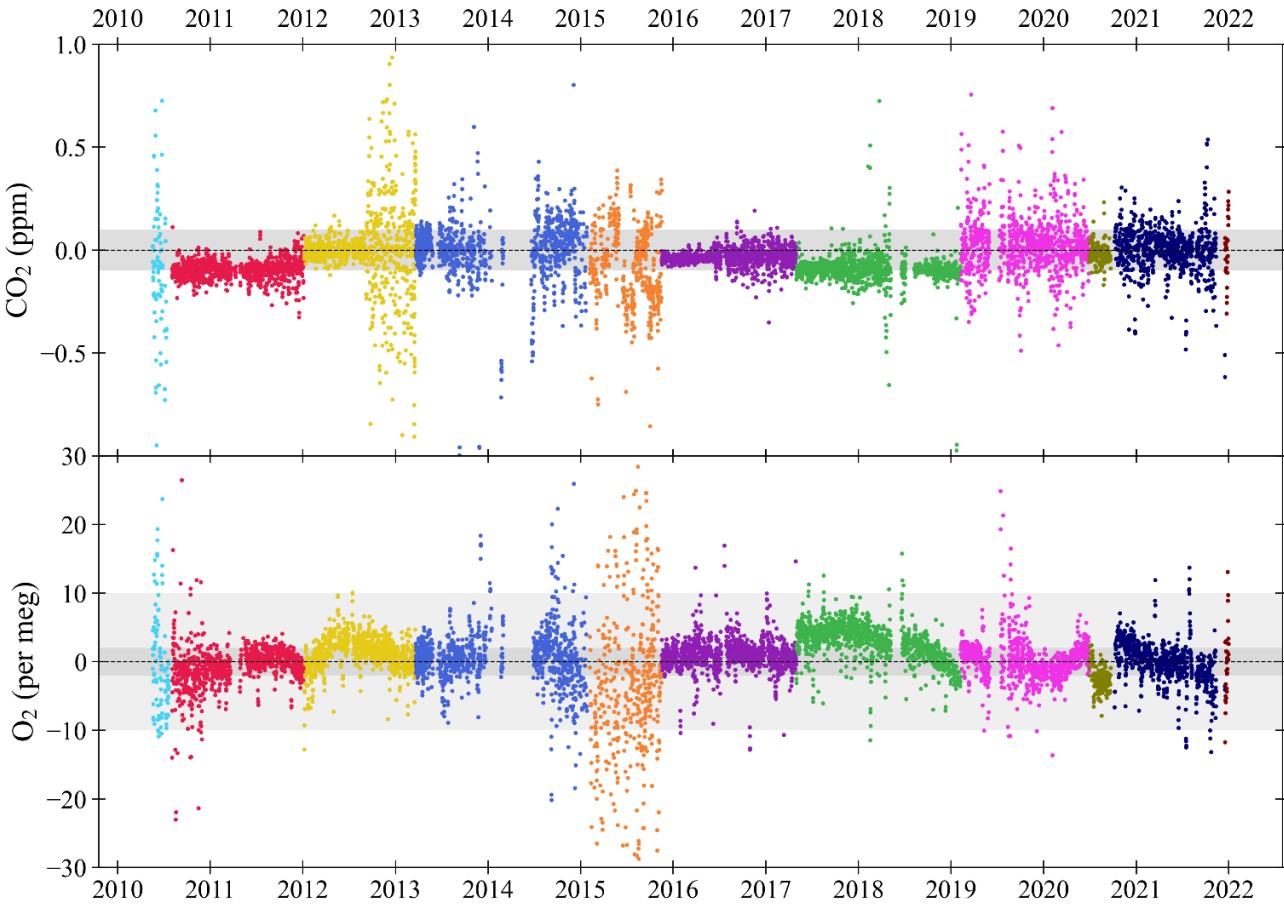

**Figure 4: Target Tank (TT) measurements of O₂ (top panel) and CO₂ (bottom panel) at Weybourne Atmospheric Observatory (WAO) between May 2010 and December 2021. Data are plotted as the difference from the CRAM Laboratory 'declared' values (measured minus declared), with each cylinder plotted in a different colour. Each TT measurement shown is the mean of 7 consecutive two-minute measurements of O₂ and CO₂. Note that 10 and 18 measurements are off scale in the O₂ and CO₂ panels, respectively, out of more than 7,000 measurements shown for each species. The O₂ y-axis is not visually comparable on a mole per mole basis to the CO₂ y-axis. Horizontal dashed lines are at zero difference, and the horizontal shaded grey bands represent the WMO compatibility goals: ±0.1 ppm for CO₂, ±2 per meg and ±10 per meg for O₂.**



**Figure 5: Linear trend lines for each Target Tank (TT), with slopes (in per meg year⁻¹ and ppm year⁻¹, for O₂ and CO₂, respectively) and R² of the Pearson correlation reported for each TT, for O₂ (top panel) and CO₂ (bottom panel) at Weybourne Atmospheric Observatory (WAO) between May 2010 and December 2021. Cylinders are numbered in the order that they were used. As in Fig. 4, horizontal dashed lines are at zero difference between WAO and UEA, and the horizontal shaded grey bands represent the WMO compatibility goals: ±0.1 ppm for CO₂, ±2 per meg and ±10 per meg for O₂. The 11ᵗʰ TT is not shown as the time period that it was in use is too short for meaningful results.**

The O₂ and CO₂ TT results as differences from the CRAM declared values are shown in Fig. 4, and in Fig. 5, for each TT cylinder, we show the linear trend lines of the TT differences with their slope and Pearson correlation ($R^2$). Ideally, both the slope of the linear trend line and the $R^2$ for each cylinder would be zero. Anything other than zero could indicate a drift in the calibration scales defined by the measurement system at WAO or a drift in the mole fractions in the TT itself.

The WAO CO₂ measurements for TT02 and TT07 are both approximately 0.1 ppm lower than the UEA CRAM declared values (Fig. 5). For CO₂ TT01 and TT05 both have decreasing trends (Fig. 5), however, these TTs exhibited more variable



measurements than the other TTs (Fig. 4), so it is possible that these trends are not robust. All the TTs, except TT01, have linear trend lines with slopes that are less than $\pm 0.15$ ppm year$^{-1}$ and $R^2$ <0.02, suggesting that there is no significant drift in $CO_2$ mole fractions. In September 2012 the TT $CO_2$ measurements were more variable due to decreased performance of the $CO_2$ analyser (Fig. 4).


For $O_2$, there are 5 cylinders that have a slope greater than $\pm 2$ per meg year$^{-1}$ (Fig. 5). TT01 has a decreasing trend and TT05 has an increasing trend, however, these TTs exhibited more variable measurements than the other TTs (Fig. 4), which may have influenced the perceived drifts in their values and they both have $R^2$ <0.09. TT01 is a 20 L cylinder (all other TTs are between 40-50 L), and there is evidence that smaller cylinders have less stable mole fractions (Pickers 2016) and this cylinder was used, with no conditioning, immediately after it had been evacuated. TT05 was the TT being used in 2015 when the Oxzilla precision was poor. TT01 and TT05 are also the cylinders with the poorest compatibility and repeatability, excluding the final cylinder (Table 1). There are three remaining cylinders that show evidence of drift and have $R^2$ >0.09. TT09 and TT10 both have decreasing $O_2$ mole fractions during the whole of their run, whereas TT07 $O_2$ mole fractions only start decreasing during the last 4 months of its run (Fig. 4). For these three cylinders the Zero Tank (ZT) $O_2$ measurements for the same time periods were investigated (see Sect. 3.3). The ZT is used to adjust the $CO_2$ calibration c-term but $O_2$ is also measured during these runs, because the analysers are connected in series. Therefore, if the ZT $O_2$ is also drifting then that would indicate that the measurement calibration is drifting, whereas if it is not drifting that would indicate that it is the mole fraction in the TT itself which is drifting. For TT07, TT09 and TT10 the ZT $O_2$ mole fractions were not drifting at the same time and therefore the drift is most likely caused by the TTs themselves.


**Table 1: Repeatability and compatibility of the WAO $O_2$ and $CO_2$ measurement system indicated by the Target Tanks (TT). Note, we report air measurements between May 2010 and December 2021, so we investigated TT measurements for the same time period, this means the first and last cylinders do not include the whole time period when they were used.**

| Target Tank | Cylinder ID | Time period | No. of TT runs | Compatibility | | Repeatability[b] | |
| --- | --- | --- | --- | --- | --- | --- | --- |
| | | | | $O_2$ (per meg) | $CO_2$ (ppm) | $O_2$ (per meg) | $CO_2$ (ppm) |
| **01** | D88528 | May 2010 - Jul 2010 | 129 | — $\pm 7.5$[a] | -0.087 $\pm$ 0.369 | $\pm 5.9 \pm 4.1$ | $\pm 0.005 \pm 0.003$ |
| **02** | D255734 | Jul 2010 - Jan 2012 | 1050 | -0.7 $\pm$ 3.1 | -0.095 $\pm$ 0.071 | $\pm 2.6 \pm 4.3$ | $\pm 0.005 \pm 0.003$ |
| **03** | D743657 | Jan 2012 - Mar 2013 | 906 | 1.2 $\pm$ 2.4 | 0.003 $\pm$ 0.216 | $\pm 1.7 \pm 1.1$ | $\pm 0.004 \pm 0.002$ |
| **04** | D743656 | Mar 2013 - Jan 2015 | 1202 | 0.6 $\pm$ 3.9 | 0.003 $\pm$ 0.151 | $\pm 3.3 \pm 5.5$ | $\pm 0.005 \pm 0.003$ |
| **05** | D273555 | Feb 2015 - Nov 2015 | 669 | -3.6 $\pm$ 11.4 | -0.065 $\pm$ 0.228 | $\pm 10.5 \pm 10.6$ | $\pm 0.006 \pm 0.003$ |
| **06** | D801298 | Nov 2015 - May 2017 | 1212 | 0.9 $\pm$ 2.3 | -0.036 $\pm$ 0.040 | $\pm 2.3 \pm 2.4$ | $\pm 0.003 \pm 0.003$ |
| **07** | D073406 | May 2017 - Feb 2019 | 1259 | 2.8 $\pm$ 2.7 | -0.093 $\pm$ 0.076 | $\pm 2.2 \pm 1.9$ | $\pm 0.005 \pm 0.043$ |
| **08** | D743656 | Feb 2019 - Jun 2020 | 982 | 0.2 $\pm$ 3.0 | 0.025 $\pm$ 0.133 | $\pm 2.8 \pm 3.7$ | $\pm 0.008 \pm 0.043$ |
| **09** | ND29112 | Jun 2020 - Sep 2020 | 152 | -2.5 $\pm$ 1.8 | -0.019 $\pm$ 0.050 | $\pm 3.0 \pm 1.6$ | $\pm 0.004 \pm 0.002$ |
| **10** | D089506 | Oct 2020 - Dec 2021 | 773 | — $\pm 3.3$[a] | — $\pm 0.119$[a] | $\pm 2.4 \pm 2.4$ | $\pm 0.004 \pm 0.002$ |





| **11** | D258964 | Dec 2021 - Dec 2021 | 30 | — ± 5.2[a] | — ± 0.144[a] | ±9.1 ± 4.6 | ±0.004 ± 0.002 |
|---|---|---|---|---|---|---|---|
| **Total** | | **May 2010 - Dec 2021** | **8365** | **0.5 ± 4.4** | **-0.037 ± 0.146** | **±3.0 ± 4.6** | **±0.005 ± 0.023** |

**[a]These cylinders were not measured at the UEA CRAM Laboratory, so the compatibility cannot be calculated, but the average of the WAO measurements was used to calculate ±1σ standard deviation for each cylinder.**
**[b]Repeatability was calculated using the same method as Kozlova and Manning (2009) and Pickers et al. (2017), from the mean ±1σ standard deviations of the average of two consecutive 2-minute TT measurements. TT repeatability is reported with ±1σ uncertainty, which recognizes the fact that the measurement system repeatability varies over time.**


Table 1 shows the compatibility of the measurement system which is the average of the differences between the TT measured values and UEA CRAM Laboratory 'declared' values and the ±1σ standard deviation. The UEA CRAM Laboratory calibration scales are traceable to the WMO $CO_2$ X2019 scale and the SIO $O_2$ scale (see Sect. 2.2). Some of the TTs were not measured in the UEA CRAM Laboratory, (T10 and T11 for both $CO_2$ and $O_2$ and TT01 for just $O_2$), so for

these cylinders we cannot calculate the compatibility, but we did use the average of the WAO measurements to calculate ±1σ standard deviation for each cylinder. The overall compatibility between 2010 and 2021 was calculated as an average, excluding these three cylinders.

The overall compatibility for $CO_2$ is -0.037 ± 0.146 ppm which is within the WMO compatibility goal (±0.1 ppm), although

the ±1σ standard deviation is slightly larger than the compatibility goal. The TT $CO_2$ compatibility is <±0.1 ppm for 67% of the measurements. The overall compatibility for $O_2$ is 0.5 ± 4.4 per meg which is within the compatibility goal of ±2 per meg, although the ±1σ standard deviation is larger than the compatibility goal. The TT $O_2$ compatibility is <±2 per meg for 54%, <±5 per meg for 87% and <±10 per meg for 96% of the measurements. For the eight TTs that we have an $O_2$ compatibility for, five of the cylinders have an average $O_2$ compatibility of <±2 per meg and the remaining three TTs have

an average $O_2$ compatibility of <±5 per meg. For all but one of the TTs the ±1σ standard deviation is within the ±10 per meg goal. The average overall compatibility is smaller than the ±1σ standard deviation for both $O_2$ and $CO_2$, meaning there is not a statistically significant difference between the WAO measurement system and the UEA CRAM Laboratory.

Ideally, the repeatability of a measurement system should be no more than half the compatibility goal, i.e. ±0.05 ppm for

$CO_2$ and ±1 per meg / ±5 per meg for $O_2$. Table 1 shows the repeatability of the TTs for each cylinder. Each TT data point is the mean of 7 consecutive 2-minute averages. We calculate the mean of ±1σ standard deviations for every two consecutive 2-minute averages for each TT run. This average of the ±1σ standard deviation for each TT run is then averaged together for each cylinder and for the whole time period to determine the repeatability. The TT repeatability is reported with the ±1σ standard deviations of the averages for each TT run.

Overall, the repeatability of the measurement system was ±0.005 ± 0.023 ppm for $CO_2$ which is an order of magnitude smaller than the repeatability goal. For $O_2$ the overall repeatability was ±3.0 ± 4.6 per meg which is greater than the ±1 per

meg ideal repeatability goal but is within the extended repeatability goal of ±5 per meg. The $CO_2$ repeatability is smaller than the $CO_2$ compatibility, -0.037 ± 0.146 ppm, whereas the $O_2$ repeatability is larger than the $O_2$ compatibility, 0.5 ± 4.4 per meg, because $O_2$ repeatability is much more technically challenging than $CO_2$ repeatability. The ±1σ standard deviations for

the $O_2$ repeatability and the $O_2$ compatibility are similar, ±4.6 per meg and ±4.4 per meg, respectively.

### 3.3 Stability of Zero Tank mole fractions

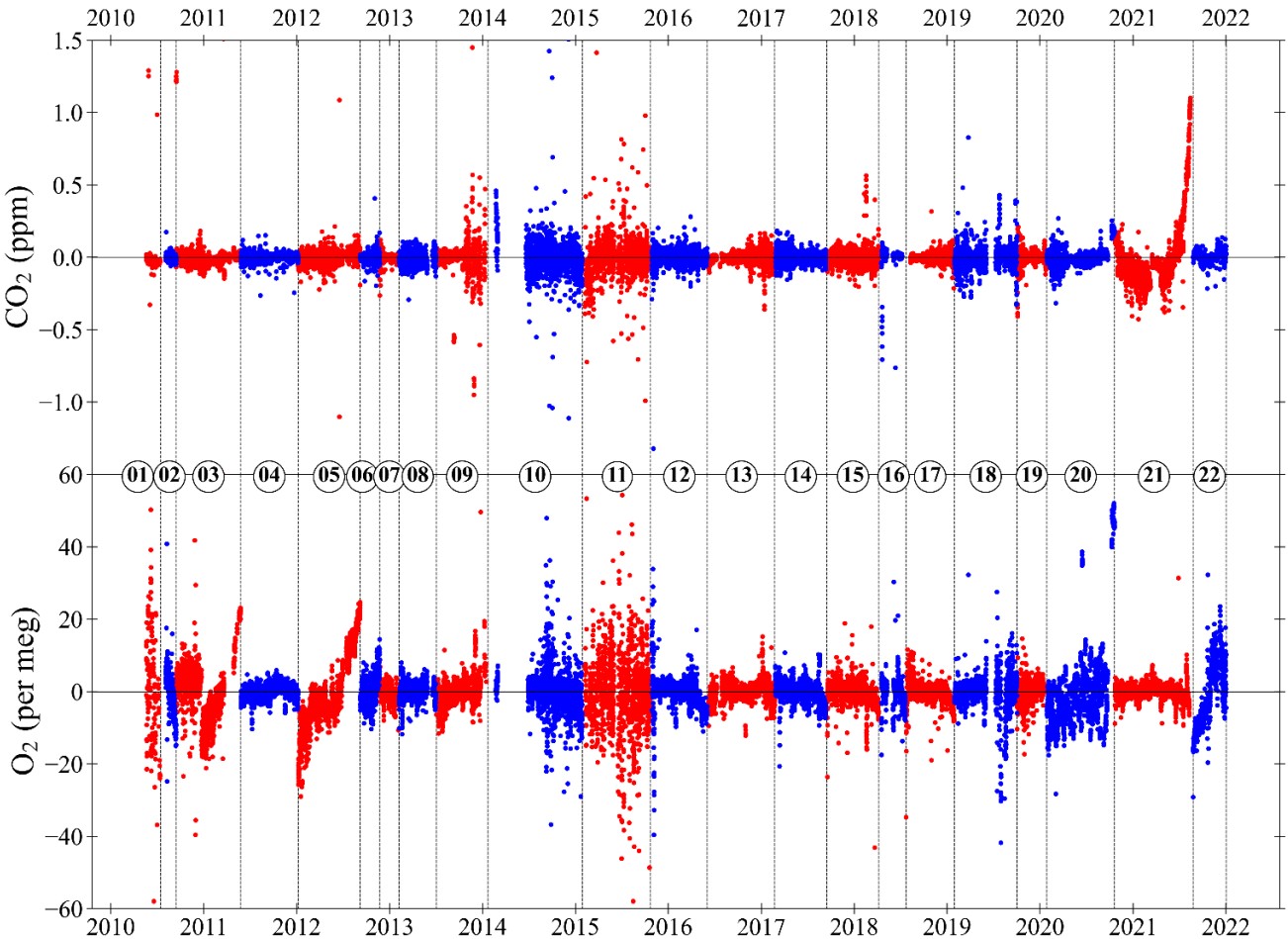

**Figure 6: The $CO_2$ (top panel) and $O_2$ (bottom panel) Zero Tank (ZT) mole fractions as the difference from the**
**average mole fraction (measurement minus average) for each ZT at WAO between May 2010 and December 2021.**
**On the y-axes 'zero' is the average mole fraction for each ZT. The $O_2$ y-axis is not visually comparable on a mole per**
**mole basis to the $CO_2$ y-axis. The ZTs alternate between red and blue to show when the ZT changes. Vertical dashed**
**lines also indicate when the ZT changes, and the cylinders are numbered in the order that they were used. Note that 7**
**and 7 measurements are off scale in the $O_2$ and $CO_2$ panels, respectively, out of more than 20,000 measurements**
**shown for each species.**

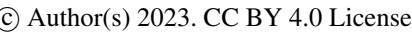

The Zero Tanks (ZTs) are used to adjust the intercept of the $CO_2$ calibration curve in-between calibrations to correct for baseline drift in the $CO_2$ analyser response caused by changes in temperature (see Sect. 2.2). The $O_2$ in the ZT is also measured but is not used as part of the calibration. The ZT mole fractions are not measured at the UEA before being sent to

WAO and therefore the ZTs cannot be used to assess the compatibility of the WAO system with the UEA CRAM Laboratory. The ZTs can be used to access the $O_2$ and $CO_2$ repeatability of the measurement system in the same way as the TTs (see Sect. 3.2).

The ZT is typically measured every 3 to 4 hours and between May 2010 and December 2021 there were 26,359 ZT

measurements. Measurements made when the system was experiencing known technical issues have been removed: this was 2,672 measurements for $CO_2$ and 2,866 measurements for $O_2$, leaving 90% of the $CO_2$ ZT data and 88% of the $O_2$ ZT data. The ZT mole fractions were between 354 ppm and 429 ppm for $CO_2$. Excluding ZT20, which had an $O_2$ mole fraction of 294 per meg, the ZT $O_2$ mole fraction ranged from -478 per meg to -1363 per meg.

The $O_2$ and $CO_2$ ZT mole fractions are shown in Fig. 6 as the difference from the average mole fraction of each cylinder. Table 2 shows the average ZT repeatability calculated in the same way as the TT repeatability (see Sect. 3.2) and the drift in the ZT $O_2$ mole fractions over time. We can see that there were time periods when the system was more stable and when it was less stable. This in general is similar to the TTs, for example, the ZTs also show that 2015 had very noisy oxygen measurements due to poor Oxzilla performance.


ZT03, ZT05 and ZT22 have $O_2$ mole fractions that drift upwards over time because they were not always stored horizontally (Fig. 6). ZT03 was originally positioned horizontally but then in December 2010 it was moved out of the Blue Box and was positioned vertically, to make space for other cylinders, and this caused the ZT $O_2$ mole fractions to start drifting upwards. ZT05 was positioned vertically for the whole of its run so its $O_2$ mole fractions are drifting upwards the whole time. ZT22

was original positioned vertically and then in November 2021 it was put horizontally. This repositioning is why the measurements for ZT22 drift upwards at the beginning and then stop. The ZT $O_2$ measurement is not used in the calibration so these drifts do not affect the measurement system performance. ZT02 has $O_2$ mole fractions that drift downwards over time. The TTs $O_2$ mole fractions were not drifting at the same time so it's likely that the ZT itself was drifting, not the measurement system. ZT01 shows a decreasing $O_2$ linear trend line, however, during this time the measurements were noisy,

so we have low confidence in the slope of the trend.

The $CO_2$ mole fraction of ZT21 drifts upwards at the end because the internal walls of this cylinder were cleaned using 'sand blasting' and it was then measured at WAO to investigate what effect this would have on the stability of the $O_2$ and $CO_2$. It was found the $CO_2$ mole fractions in the cylinder drift upwards at lower pressures, but there was no noticable effect on $O_2$.



The ZT is used to recalibrate the intercept every 3 to 4 hours, and the within-cylinder drift in this time period should be small enough to not affect the measurements. In order to be confident that there is no effect on the measurements, 'sandblasted' cylinders should not be used as ZTs once the internal pressure drops below 15 bar.

If we exclude the 3 ZTs that were vertical, the ZT repeatability is on average for $O_2$ ±3.1 ± 5.3 per meg, which is more than 475 the ±1 per meg repeatability goal, and less than the extended repeatability goal of ±5 per meg, but the ±1σ standard deviation is higher. The $O_2$ repeatability based on the ZTs is similar to the $O_2$ repeatability based on the TTs (±3.0 ± 4.6 per meg, see Sect. 3.2), which increases our confidence, that this is the $O_2$ repeatability of the measurement system. The average $CO_2$ repeatability based on the ZTs is ±0.005 ± 0.019 ppm, which is again similar to the $CO_2$ repeatability based on the TTs, (±0.005 ± 0.023 ppm, see Sect. 3.2).


**Table 2: Repeatability of the Zero Tank (ZT) $O_2$ and $CO_2$ measurements at WAO. We have calculated the $O_2$ linear trend line for the time period each ZT was used and report here the slope and $R^2$ of the Pearson correlation. Note, we report air measurements between May 2010 and December 2021, so we investigated ZT measurements for the same time period, this means the first and last cylinders do not include the whole time period when they were used.**

| Zero Tank | Time period | No. of ZT runs | $O_2$ slope (per meg year$^{-1}$) | $O_2$ $R^2$ | $O_2$ repeatability (per meg) | $CO_2$ repeatability (ppm) |
|---|---|---|---|---|---|---|
| 01 | May 2010 - Jul 2010 | 220 | -127.5 | 0.099 | ±7.0 ± 6.7 | ±0.004 ± 0.002 |
| 02 | Jul 2010 - Sep 2010 | 306 | -150.0 | 0.373 | ±5.6 ± 9.5 | ±0.005 ± 0.003 |
| 03 | Sep 2010 - May 2011 | 1407 | 1.3 | 0.001 | ±3.7 ± 5.0 | ±0.006 ± 0.023 |
| 04 | May 2011 - Jan 2012 | 1276 | 3.0 | 0.076 | ±1.5 ± 0.8 | ±0.005 ± 0.003 |
| 05 | Jan 2012 - Sep 2012 | 1323 | 46.4 | 0.791 | ±1.7 ± 0.8 | ±0.005 ± 0.003 |
| 06 | Sep 2012 - Nov 2012 | 453 | 29.8 | 0.199 | ±1.8 ± 0.9 | ±0.003 ± 0.002 |
| 07 | Nov 2012 - Feb 2013 | 948 | -16.2 | 0.149 | ±1.8 ± 0.9 | ±0.003 ± 0.001 |
| 08 | Feb 2013 - Jul 2013 | 1040 | 1.2 | 0.003 | ±1.8 ± 1.3 | ±0.004 ± 0.002 |
| 09 | Jul 2013 - Jan 2014 | 1241 | 13.3 | 0.235 | ±2.3 ± 1.7 | ±0.006 ± 0.031 |
| 10 | Jan 2014 - Jan 2015 | 1760 | -7.1 | 0.058 | ±4.9 ± 8.3 | ±0.005 ± 0.005 |
| 11 | Feb 2015 - Oct 2015 | 1903 | -0.9 | 0.000 | ±10.8 ± 12.3 | ±0.006 ± 0.003 |
| 12 | Oct 2015 - Jun 2016 | 1711 | -7.7 | 0.089 | ±3.4 ± 7.2 | ±0.003 ± 0.001 |
| 13 | Jun 2016 - Feb 2017 | 1624 | 1.1 | 0.007 | ±2.2 ± 1.7 | ±0.004 ± 0.002 |
| 14 | Feb 2017 - Sep 2017 | 1588 | -6.6 | 0.155 | ±2.2 ± 1.7 | ±0.003 ± 0.002 |
| 15 | Sep 2017 - Dec 2017 | 1555 | -5.7 | 0.086 | ±2.5 ± 3.2 | ±0.006 ± 0.029 |
| 16 | Jun 2018 - Jul 2018 | 636 | -10.2 | 0.047 | ±2.5 ± 2.8 | ±0.006 ± 0.054 |
| 17 | Jul 2018 - Jan 2019 | 1401 | -9.2 | 0.211 | ±2.0 ± 2.0 | ±0.003 ± 0.009 |
| 18 | Jan 2019 - Oct 2019 | 1502 | -0.1 | 0.000 | ±3.5 ± 5.3 | ±0.010 ± 0.049 |
| 19 | Oct 2019 - Jan 2020 | 861 | 10.6 | 0.092 | ±3.0 ± 3.2 | ±0.004 ± 0.002 |
| 20 | Jan 2020 - Oct 2020 | 1434 | 32.9 | 0.295 | ±2.2 ± 1.5 | ±0.003 ± 0.001 |
| 21 | Oct 2020 - Aug 2021 | 1575 | -0.6 | 0.005 | ±1.8 ± 1.9 | ±0.003 ± 0.002 |
| 22 | Aug 2021 - Dec 2021 | 595 | 65.8 | 0.566 | ±5.7 ± 5.2 | ±0.004 ± 0.013 |




Figure 7 shows the absolute difference in the $CO_2$ mole fractions between the current ZT and the previous ZT (measured 3 to 4 hours earlier). The ZT is used to adjust the intercept of the $CO_2$ calibration curve in-between calibrations and it is assumed that the difference between the current ZT and the previous ZT is caused by drift in the baseline response of the $CO_2$ analyser. The average absolute difference in $CO_2$ is 0.03 ppm which is less than the ±0.05 ppm repeatability goal, overall, 85% of the differences are <0.05 ppm. These differences suggest that the ZT is measured regularly enough that the drift in the baseline is kept small enough to not substantially effect the measurements.

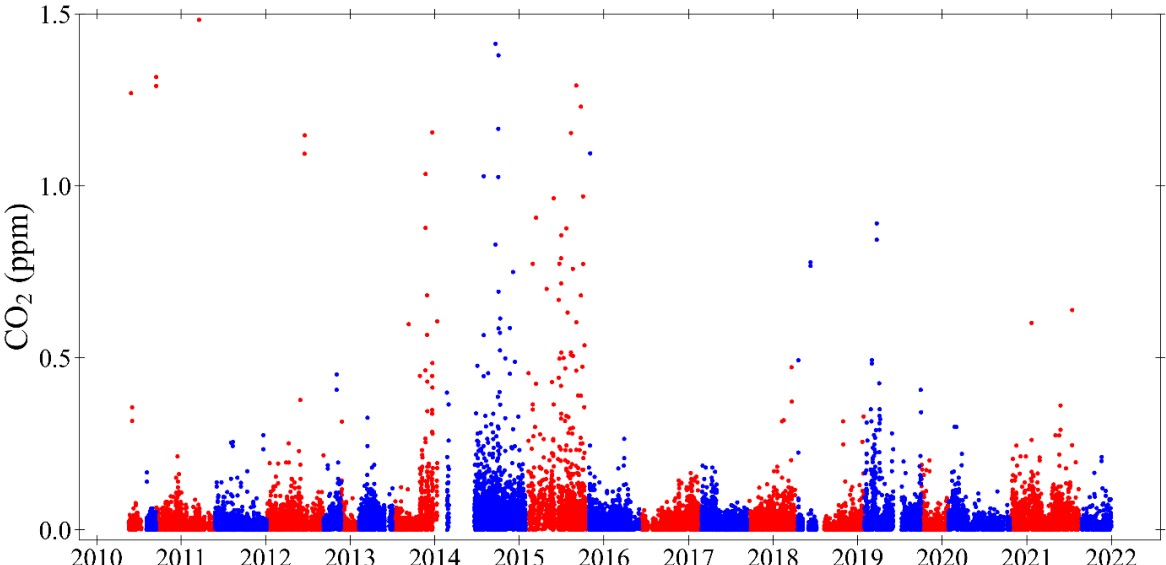

**Figure 7: Absolute differences between the ZT $CO_2$ mole fraction and the previous ZT $CO_2$ mole fraction (current minus previous; typically measured 3 to 4 hours earlier). The red and blue alternating colours show when the ZT changes. Note that 13 measurements are off scale, out of more than 20,000 measurements shown.**

## 3.4 Stability of calibration coefficients



**Figure 8: The $CO_2$ a and b terms, from the $CO_2$ calibration quadratic equation, $y = ax^2 + bx + c$, and the $O_2$ b-term**
**and $O_2$ $R^2$, from the $O_2$ calibration linear equation, $y = bx + c$, of the WSS calibrations at WAO between May 2010**
**and December 2021. The colours indicate different WSS Sets of cylinders (Set 1 = red and pink, Set 2 = blue, Set 3 =**
**green) and 1.0 on the $O_2$ $R^2$ panel is indicated by a horizontal orange line. The c-terms for each species, equivalent to**
**the WT mole fractions, are shown in Fig. 10. Note that 5 measurements are off scale in the $O_2$ $R^2$ panel, out of more**
**than 1700 measurements shown.**


Three Working Secondary Standard (WSS) cylinders are used to calibrate the measurement system every 47 hours (see Sect. 2.2). The c-terms of these equations are discussed in Sect. 3.5. In this section we discuss the a-term and b-terms of the $CO_2$ and $O_2$ calibrations. We also discuss the Pearson's correlation coefficient, $R^2$, of the measured $O_2$ mole fractions. Each of these parameters is shown in Fig. 8 as a time series. As three WSSes are used, and the $CO_2$ response is a quadratic equation

the $CO_2$ $R^2$ is always exactly 1.0. The Ultramat and Oxzilla report uncalibrated raw values of $CO_2$ and $O_2$ in units of vpm and %, respectively. The calibration coefficient units were converted from ppm/vpm to ppm for $CO_2$ and ppmEq/% to ppmEq for $O_2$. This conversion was done by multiplying the coefficients by the average analyser response of WSSes between June 2014 and September 2020: D255734 for $CO_2$ (399.37 ppm, -18.77 vpm) and D743654 for $O_2$ (-709.0 per meg, -122.7 ppmEq, 0.00072%). At WAO between May 2010 and December 2021, three sets of WSS cylinders were used (Table

3). In total, there were 2038 calibrations, calibrations when the system was experiencing known technical issues were excluded, this was 277 for $CO_2$ and 293 for $O_2$, so this leaves 86% of the data.

If the calibration coefficient terms drift over time, this could indicate a drift in the analyser sensitivity or a drift in the calibration scale. A drift in the sensitivity of the analyser, may impact the precision and the accuracy. By calibrating

frequently, the accuracy of the measurement should be less susceptible to drift in the analysers' sensitivity. However, a deterioration in the precision cannot be corrected for. A drift in the calibration scale may be caused by internal drift of the mole fractions in the calibration cylinders. This can only be determined by reanalysing the calibration cylinders after they have finished being measured at WAO. However, due to practical constraints this is not done on the calibration cylinders used at WAO.


There are sometimes step changes in the calibration coefficients when the WSS cylinders are changed as the cylinders have different mole fractions in them. There are also sometimes step changes when the cylinders are not changed. These step changes are most likely caused by changes in the analyser response as it is very unlikely that the mole fractions in the cylinders would suddenly change very quickly and then stop. For example, the $CO_2$ b-term has a step change in October

2013, when the Ultramat analyser was changed.

The drift in the calibration coefficients was calculated using the slope in the linear trend line for each of the five time periods WSSes were measured for (Table 3). The $CO_2$ a-term is small and therefore the drift in the $CO_2$ a-term is also small, less





than ±4 ppt year$^{-1}$, indicating that the $CO_2$ analysers at WAO have a very linear response. There is a large drift between May-

2010 and Nov-2010, the first time ND29112 (Set 1) was used, for both the $CO_2$ b-term (4.66 ppm year$^{-1}$) and $O_2$ b-term (-3.6 per meg year$^{-1}$). The smallest drift in the $CO_2$ b-term, -0.01 ppm year$^{-1}$, occurs between Jun-2014 and Sep-2020 when Set 2 of the WSSes is used. For the $O_2$ b-term, three out of the five periods have drift that is less than ±0.3 per meg year$^{-1}$. The largest drift in the $O_2$ b-term, 4.1 per meg year$^{-1}$, occurs when Set 2 of the WSSes is used, but this is strongly influenced by the calibrations in 2014 and 2015 when there were issues with the measurement system. If the slope is recalculated starting

from Jan-2016 instead, then the drift decreases to -1.9 per meg year$^{-1}$. The $O_2$ and $CO_2$ mole fractions in the Target Tanks and Zero Tanks were not drifting at the same time (see Sects. 3.2 and 3.3), therefore indicating that it is not the cylinders that are drifting, but the analyser sensitivity. The analyser sensitivity drifting should not affect the measurement results so long as the analyser response drift is not significant within a 47-hour time period.

**Table 3: All the cylinders that were used as WSSes at WAO. Three sets of WSS cylinders were used (Set 1: May 2010 - March 2014, Set 2: June 2014 - September 2020, Set 3: October 2020 - December 2021). In November 2010, ND29112 was replaced with D88531, then in May 2012 was changed back to ND29112. For each calibration parameter, we report the slope and $R^2$ of the linear trend line. The slopes of the $CO_2$ a-terms are in units of ppt year$^{-1}$ (parts per trillion per year). The slopes of the $O_2$ b-terms were multiplied by 6.05 to convert them from ppmEq year$^{-1}$**

**to per meg year$^{-1}$.**

| Set | Cylinder IDs | Start Date | End Date | $CO_2$ a-term | | $CO_2$ b-term | | $O_2$ b-term | | |
|---|---|---|---|---|---|---|---|---|---|---|
| | | | | Slope (ppt year$^{-1}$) | $R^2$ | Slope (ppm year$^{-1}$) | $R^2$ | Slope (ppmEq year$^{-1}$) | Slope (per meg year$^{-1}$) | $R^2$ |
| 1 | ND29112 ND29108 ND29109 | May-2010 | Nov-2010 | -3.27 | 0.29 | -4.66 | 0.82 | -0.59 | -3.6 | 0.68 |
| 1 | D88531 ND29108 ND29109 | Nov-2010 | May-2012 | 0.92 | 0.19 | 0.62 | 0.44 | -0.01 | -0.1 | 0.03 |
| 1 | ND29112 ND29108 ND29109 | May-2012 | Mar-2014 | -1.09 | 0.08 | 0.79 | 0.07 | -0.04 | -0.3 | 0.20 |
| 2 | D743654 D255697 D255734 | Jun-2014 | Sep-2020 | 0.04 | 0.02 | -0.01 | 0.00 | 0.68 | 4.1 | 0.20 |
| 3 | D743657 D269489 D269491 | Oct-2020 | Dec-2021 | 2.54 | 0.05 | 0.42 | 0.09 | -0.03 | -0.2 | 0.06 |

The $O_2$ $R_2$ varies over time. It is highest and most stable when Set 1 of the WSSes is being measured between 2010 and 2014, on average the $O_2$ $R^2$ during this time is 0.9997. In 2014 and 2015 there is additional evidence of the poorer performance of the Oxzilla analyser during this time, and the $O_2$ $R^2$ are smaller and much more variable. From 2016 to 2020

the $O_2$ $R^2$ is higher and more stable, although it is not as good as it was when Set 1 of the WSSes were being measured, on average the $O_2$ $R^2$ during this time is 0.9968. When Set 3 of the WSSes is being measured the $O_2$ $R^2$ is higher and more





stable than it was with Set 2 but still not as good as it was with Set 1, on average for Set 3 the $O_2$ $R_2$ is 0.9988. Overall, from May-2010 to Dec-2021, the $O_2$ $R^2$ is on average 0.9977 and is >0.995, 92% of the time.

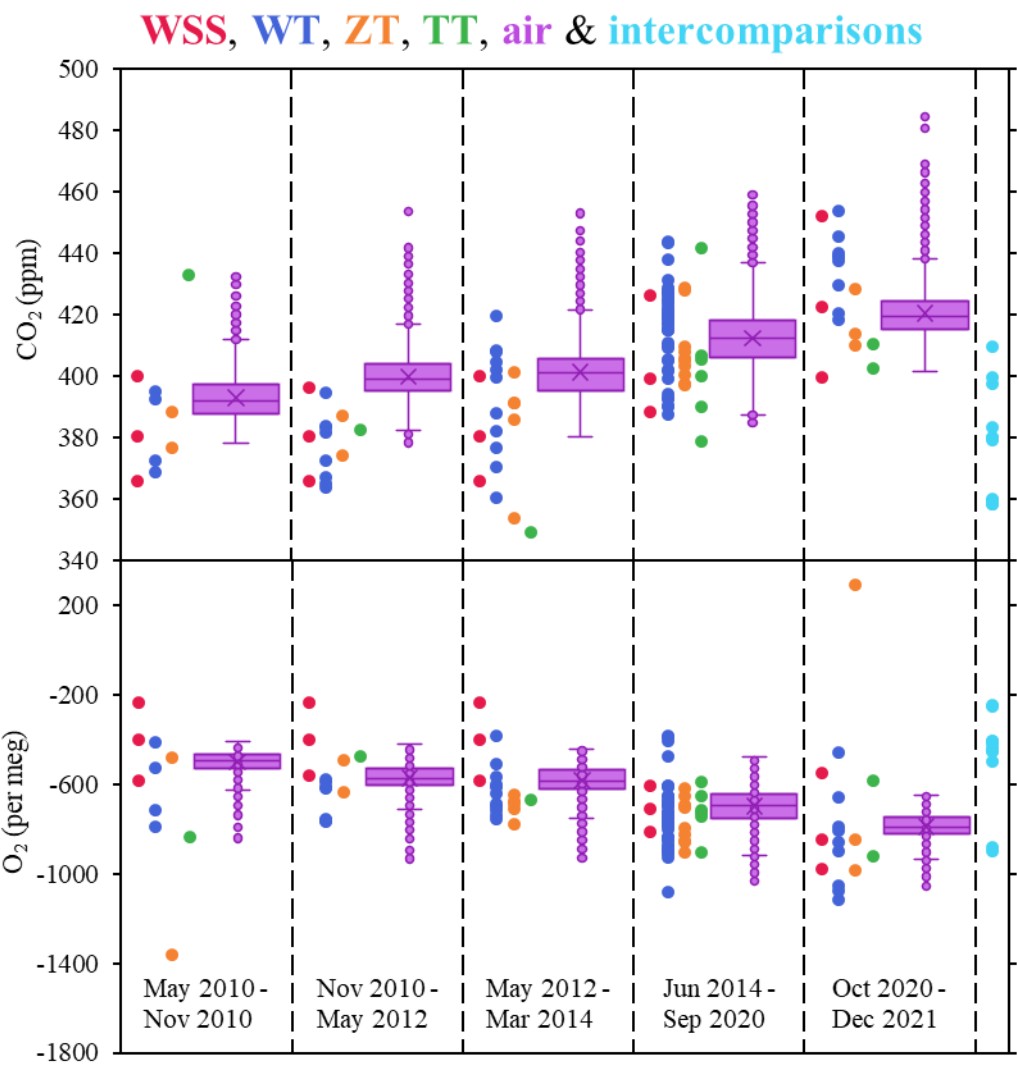

**Figure 9: The CO₂ (top panel) and O₂ (bottom panel) mole fractions of the WSSes, WTs, ZTs, TTs and intercomparison cylinders measured at WAO between May 2010 and December 2021, colour coded as indicated by the title at the top of the figure. The air measurements made at WAO during this time are shown as box and whisker plots.**

The calibration range is between the WSS cylinder with the highest mole fraction and the WSS cylinder with the lowest mole fraction. It is intended that the calibration range is large enough to cover the range of mole fractions likely to be

measured in the air at WAO. For $CO_2$, 77% of the air measurements, 54 out of 73 WTs, 18 out of 22 ZTs and 7 out of 11 TTs are within the calibration range (Fig. 9). For $O_2$, 74% of the air measurements, 31 out of 73 WTs, 9 out of 22 ZTs, and 7

out of 11 TTs, are within the calibration range (Fig. 9). We can have less confidence in the accuracy of mole fractions measured outside the calibration range, and the further away from our calibration range the measurement is, the less confidence we can have. This is more the case for $CO_2$, for which the analyser has a non-linear response, than for $O_2$, which has a linear response.

## 3.5 Stability of Working Tank mole fractions


**Figure 10: The CO₂ (top panel) and O₂ (bottom panel) Working Tank (WT) mole fractions calculated from the calibrations as the difference from the average mole fraction for each WT (measurement minus average) at WAO between May 2010 and December 2021. The WTs alternate between blue and red to show when the WT changes. On**

**the y-axes 'zero' is the average mole fraction for each WT (dashed line). The O₂ y-axis is not visually comparable on a**



**mole per mole basis to the CO₂ y-axis. Note that 5 and 7 measurements are off scale in the O₂ and CO₂ panels, respectively, out of more than 1700 measurements shown for each species.**

Since all measurements are taken against the reference of the Working Tank (WT) this means that the intercept (i.e., the c-term) of the calibration curves for both $CO_2$ and $O_2$ is the WT mole fraction at the point of calibration. Calibrations typically take place every 47 hours and the stability of the WT mole fraction over time provides another measure of system performance. Usually, WTs are 50L cylinders filled to ~200 bar, which are used until their pressure gets below ~5 bar. The measurement system flow rate is approximately 100 ml/minute and on average WTs last 55 days. Between May 2010 and

December 2021, 76 Working Tanks were measured at WAO. In total there were 2038 calibrations between May 2010 and December 2021. Calibrations made when the system was experiencing known technical issues have been excluded. This leaves 1761 $CO_2$ calibrations and 1747 $O_2$ calibrations, or in other words 86% of the calibrations. WT mole fractions for $CO_2$ were between 360 ppm and 454 ppm and for $O_2$ were between -1115 per meg and -362 per meg.

Figure 10 shows the difference between the current WT mole fraction and the average WT mole fraction for each cylinder over its lifetime, as defined by the WSS calibrations. In 2014 and 2015 the WTs mole fractions were much more unstable than they typically are due to multiple technical issues (see Sect. 4.1). The absolute difference of the WT mole fraction between the current calibration and the previous calibration was on average 2.1 ± 3.3 per meg for $O_2$. This difference is above the WMO repeatability goal of ±1 per meg, but below the extended repeatability goal of ±5 per meg. For $O_2$, the

difference between successive calibrations indicates whether the calibration cycle is being run often enough to prevent significant baseline drift. For $CO_2$, the absolute difference between calibrations is 0.08 ± 0.38 ppm, which is above the repeatability goal of ±0.05 ppm, however, this does not matter since the WT is redefined every time the ZT is run (every 3 to 4 hours), so this merely confirms the need to run the ZT every 3 to 4 hours in order to correct for $CO_2$ baseline drift between calibrations.


The $O_2$ mole fraction decreases as the air in the cylinder is consumed (Figs. 10 and 11). This effect is most likely caused by the preferential desorption of $N_2$ relative to $O_2$ from the interior walls of the cylinders as the pressure decreases, owing to the difference in their molecular mass. This effect has been previously observed in other studies (Keeling et al., 1998; Manning 2001; Keeling et al., 2007; Kozlova and Manning 2009; Wilson 2012; Pickers 2016; Barningham 2018). On average for

every 1 bar used $O_2$ decreases by 0.034 per meg (Fig. 11), assuming a WT starts at 200 bar and finishes at 5 bar this is a decrease of 6.6 per meg over the lifetime of the cylinder. Previous studies have found various depletion amounts: 5 per meg (Keeling et al., 1998), 6 per meg (Manning 2001), 1 per meg (Keeling et al., 2007), 30 per meg (Kozlova and Manning 2009), 12 per meg (Wilson 2012), 54 per meg (Pickers 2016), 6.6 per meg (Barningham 2018). Some of these differences can be explained by differences in pressures and flow rates between the different measurement systems (Barningham 2018),





or using different types of cylinders (Pickers, 2016). The WT $CO_2$ does not change as the pressure decreases. The $O_2$ depletion is not an issue for WTs as their mole fractions are redefined after every WSS calibration (i.e., every 47 hours).

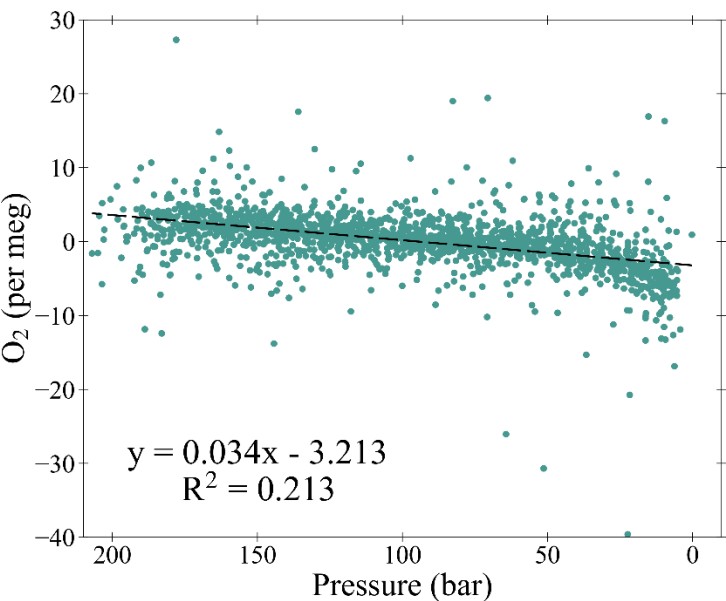

**Figure 11: An aggregate of all the $O_2$ Working Tank (WT) mole fractions determined during all of the calibrations, plotted as the difference from the average mole fraction for each WT (measurement minus average) at WAO between May 2010 and December 2021, plotted versus the WT pressure. The linear trend line is denoted as a dashed black line.**



## 3.6 Standard deviation of the air measurements

Figure 12: CO₂ (top panel) and O₂ (bottom panel) ±1σ standard deviation of air measurements at WAO between May 2010 and December 2021. The ±1σ standard deviations of the 2-minute data were not routinely calculated for O₂ until February 2012. Hourly averages of the ±1σ standard deviations of the 2-minute measurements. The O₂ y-axis is not visually comparable on a mole per mole basis to the CO₂ y-axis. Note that 4 measurements are off scale in the CO₂ panel, out of more than 70,000 measurements shown.

Figure 12 shows the hourly averages of the ±1σ standard deviation of the 2-minute measurements. For each 2-minute measurement there is a ±1σ standard deviation, which is determined from the 1-minute switching of WT and sample air, using 1-second data. These ±1σ standard deviations for each hour (30 data points if there are no gaps) are then used to calculate the mean. These ±1σ standard deviations include both the uncertainty in the measurement system and how much the actual CO₂ and O₂ mole fractions in the air vary over a 2-minute period. Therefore, these ±1σ standard deviations cannot be compared to the ±1σ standard deviations of the Target Tanks or to the repeatability goals.





The ±1σ standard deviation of the $CO_2$ measurements is very stable over time (Fig. 12). On average the $CO_2$ ±1σ standard deviation is ±0.08 ± 0.10 ppm and is <0.2 ppm 93% of the time. On average the $O_2$ ±1σ standard deviation is ±6.8 ± 5.4 per meg and is <10 per meg, 94% of the time. In 2014 and 2015 the $O_2$ ±1σ standard deviation were much higher and more variable than usual, due to multiple different technical problems in 2014 and poorer performance of the Oxzilla analyser in 2015 (see Sect. 4.1). The 2014 and 2015 $O_2$ measurements are still accurate, as far as we can tell, but they are not as precise. In general, larger $O_2$ ±1σ standard deviations correspond to poorer performance of the Oxzilla analyser.

# 4 Key features of the WAO atmospheric $CO_2$, $O_2$ and APO data

## 4.1 Time series of $CO_2$, $O_2$ and APO

**Figure 13: Hourly averaged $CO_2$ (top panel), $O_2$ (middle panel) and APO (bottom panel) at WAO between May 2010 and December 2021. The baselines (yellow lines) are fitted using the RFbaseline function of Ruckstuhl et al. (2012). The curve fits (black lines) are calculated using the STL function in R (see Sect. 2.3). The $O_2$ and $CO_2$ y-axes are**



scaled to be visually comparable on a mole per mole basis. The APO y-axis is zoomed in 2 times, compared to the $O_2$ y-axis on a mole per mole basis. The x-axis tick marks are at the beginning of each year.

Figure 13 shows a 12-year time series from May 2010 to December 2021 of continuous in situ measurements of $CO_2$, $O_2$ and
APO at WAO. Measurements continue to the present and are recorded every 2 minutes. There are gaps in the data caused by cylinder measurements (see Sect. 2.2), changing the WT, routine maintenance, and technical issues. Data collected when the system was experiencing known technical issues have been removed. In total there are 1,930,178 $CO_2$ measurements and 1,565,908 $O_2$ measurements (2-minute frequency), from 19[th] May 2010 to 31[st] December 2021. The data have been averaged to hours to make the dataset more useful for typical applications. Hourly averages are only calculated when there are at least
five 2-minute measurements in an hour. There are 73,232 $CO_2$ hourly averages and 69,044 $O_2$ hourly averages meaning there are hourly measurements for 72% ($CO_2$) and 68% ($O_2$) of the time. Simultaneous measurements of both $O_2$ and $CO_2$ are needed to calculate APO.

The proportion of APO data missing in each year varies (Table 4). Measuring the WSSes, TTs and ZTs takes approximately 14% of the time in a year, so even if the measurement system ran perfectly, we would still not be able to measure air 100%
of the time. The most complete year was 2017 with only 15% of the data missing. The year with the most missing data is 2014, where more than half (59%) of the data are missing, as there were multiple technical issues in 2014. The problems in 2014 included: problems with the compressor that opened and closed the pneumatic valves, the pneumatic valves were replaced with solenoid valves in June 2014; in January/February there was no Ultramat signal because of a broken serial lead, and there was also no ZT; in May 2014 the WSS cylinders were changed; throughout the year there were electrical
problems including with the USB hub, data acquisition boards and watchdog board; in June 2014 the red line inlet was blocked with sea salt; there was also a problem with the computer frequently crashing; the PC was replaced in November 2014. The year 2015 also has a large amount of missing data, 42%, as this year also had technical issues and a different Oxzilla analyser was used in this year which was not as precise. The years 2018 and 2019 also have a large amount of missing data, 46% and 47% respectively. This is because between March 2018 to March 2019 the diaphragm in the KNF
pump was torn causing the pump to leak. During this time only one of the inlet lines was being used so this leak affected all the data. Some of these data, from March 2018 to October 2018, were adjusted but some of the data, from November 2018 to March 2019, was not so it had to be removed from the dataset. For more information on this leaking pump adjustment see the supplement. All the other years have between 20% and 33% of the data missing (Table 4).

**Table 4: Annual mean of $CO_2$, $O_2$ and APO data at WAO from May 2010 to December 2021. The percentage of APO**
**data missing each year based on the hourly averages is also shown (the APO calculation requires simultaneous $CO_2$ and $O_2$ measurements).**

| Year | CO₂ (ppm) | O₂ (per meg) | APO (per meg) | Missing data (%) |
|------|-----------|--------------|---------------|------------------|
| 2010 | 388.11 | -477.9 | -265.5 | 30% |
| 2011 | 389.88 | -494.6 | -273.0 | 30% |



| | | | | |
|---|---|---|---|---|
| **2012** | 391.97 | -516.7 | -283.8 | 23% |
| **2013** | 394.44 | -536.4 | -292.8 | 33% |
| **2014** | 396.74 | -554.8 | -303.8 | 59% |
| **2015** | 398.85 | -579.4 | -313.2 | 42% |
| **2016** | 401.53 | -609.6 | -327.3 | 29% |
| **2017** | 403.98 | -636.8 | -342.4 | 15% |
| **2018** | 406.30 | -666.1 | -357.6 | 46% |
| **2019** | 408.80 | -688.8 | -368.7 | 47% |
| **2020** | 411.57 | -709.7 | -375.1 | 20% |
| **2021** | 413.86 | -731.0 | -383.8 | 24% |

Most of the gaps in the data only last for short periods of time: 9% of the missing data are due to gaps lasting 1 hour and are usually caused by routine calibrations; 14% of the missing data are due to gaps lasting 2 hours; 26% of the missing data are due to gaps lasting less than a day; 47% of the missing data are due to gaps lasting less than a week; and 66% of the missing data are due to gaps lasting less than a month (Table 5). There are 6 gaps in the data that last longer than a month. The largest gap in the data is 5 months long and is it due to the data that had to be removed when the KNF pump was leaking between November 2018 and March 2019. The next longest gap is 4 months long and occurred in 2014 between 2[nd] March and 28[th] June due to the combination of technical problems described above.

**Table 5: Duration and number of gaps in the dataset, based on APO data.**

| Duration | No. of gaps | Percent |
|---|---|---|
| **1 hour** | 2619 | 9% |
| 2 hours | 685 | 14% |
| 3 hours - 1 day | 575 | 26% |
| 1 day - 1 week | 98 | 47% |
| 1 week - 1 month | 19 | 66% |
| >1 month | 6 | 100% |

From Fig. 13 we can see many interesting features of this dataset. Firstly, there are long-term trends in the measurements with increasing mole fractions of $CO_2$ and decreasing mole fractions of $O_2$ and APO, predominantly due to the burning of fossil fuels and land-use changes which release $CO_2$ and consume $O_2$ (Sect. 4.2). These trends are buffered by an increasing land carbon sink, and the ocean carbon sink is also buffering the $CO_2$ trend but not the $O_2$ trend (Keeling et al., 1996a). Secondly, there are seasonal cycles: $CO_2$ has higher mole fractions during the winter and lower mole fractions during the summer, while the seasonal cycles of $O_2$ and APO are reversed, with lower mole fractions during the winter and higher mole fractions during the summer (Sect. 4.3). These seasonal cycles are caused by seasonal changes in terrestrial biosphere processes, ocean processes and fossil fuel emissions. Thirdly, it can also be seen that there is a lot of synoptic variability in



the species measured at WAO, due to the influence of more local emissions within the atmospheric footprint. The spikes are less common in the APO time series therefore indicating that most of this variability is mostly from terrestrial biosphere processes, and not from the ocean.

## 4.2 Interannual variability of long-term trends

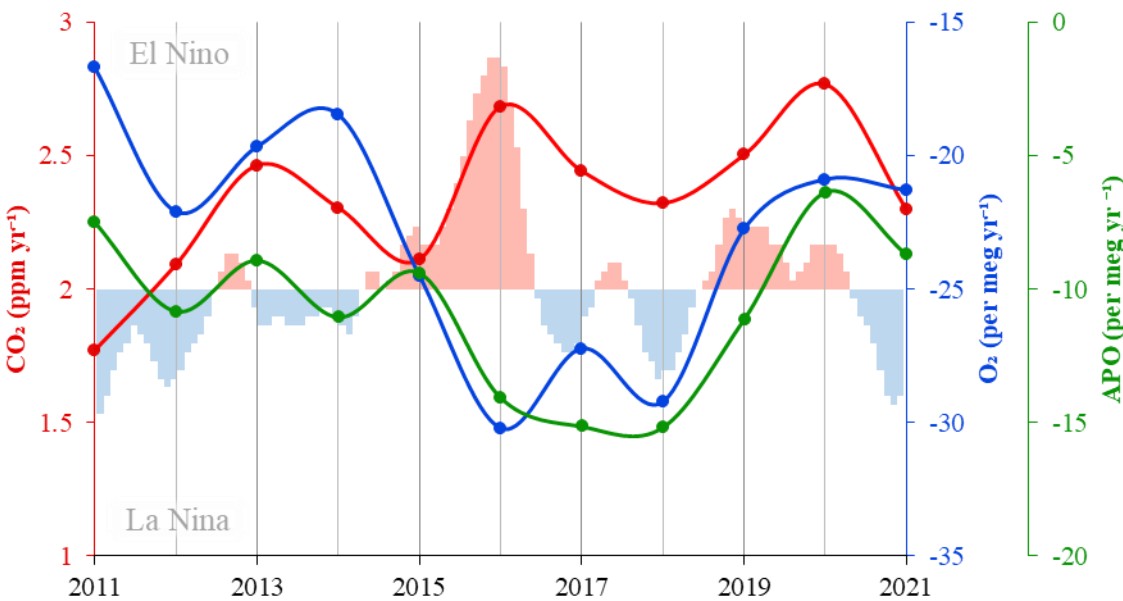


**Figure 14: Interannual variability of long-term trends in $CO_2$ (red), $O_2$ (blue) and APO (green) calculated using the difference between annual mean mole fractions at WAO from 2010 to 2021. The $O_2$ and APO y-axes are not visually comparable on a mole per mole basis to the $CO_2$ y-axis. The shaded bar chart shows the Oceanic Niño Index (ONI) that is used for identifying El Niño (red/warm) and La Niña (blue/cool) events (NOAA, 2023). Note that the values shown for 2011 use an incomplete year (7 months) for the 2010 annual means; the data were deseasonalised to prevent bias in the calculations.**


The long-term trends were calculated using the trend from the seasonal decomposition with STL (see Sect. 2.3) and the TheilSen function in the openair package in R (Carslaw and Ropkins, 2012). Overall, atmospheric $CO_2$ mole fraction at

WAO grew by 2.40 ppm yr$^{-1}$ (2.38 to 2.42). Atmospheric $O_2$ declined by 24.0 per meg yr$^{-1}$ (24.3 to 23.8). APO was observed to have an annual decrease at WAO of 11.4 per meg yr$^{-1}$ (11.7 to 11.3). Atmospheric $O_2$ is decreasing more quickly than $CO_2$ is increasing because $CO_2$ has two sinks, the land and ocean, whereas $O_2$ only has a source from the land (Keeling et al., 1996a). The average $CO_2$ growth rate at WAO is similar to the global $CO_2$ growth rate for 2012-2022 reported in the Global Carbon Budget of 2.4 ± 0.1 ppm (Dlugokencky and Tans 2022; Friedlingstein et al., 2022). The average $O_2$ trend at WAO is

slightly larger than the $O_2$ trends reported at Lutjewad, -21.2 ± 0.8 per meg yr$^{-1}$, and Mace Head, -21.3 ± 0.9 per meg yr$^{-1}$, between 2002 and 2018 (Nguyen et al., 2022).

The differences between the annual means of the trends from STL were used to examine the interannual variability of the long-term trends. While there has been a consistent overall trend, there has also been variation in the annual trends of $CO_2$ (1.8 to 2.8 ppm $yr^{-1}$), $O_2$ (-30 to -17 per meg $yr^{-1}$) and APO (-15 to -6 per meg $yr^{-1}$) (Fig. 14). The APO trend is correlated

with the $O_2$ trend (Pearson's $R^2 = 0.66$, p-value = 0.02). The interannual variability in the long-term trends of $CO_2$ and $O_2$ do not appear to be correlated or anti-correlated. There are many factors that could influence the trends: changes in northern hemispheric $CO_2$ emissions, variability in atmospheric transport, and features that lead to variations in temperature and precipitation as this will effect the terrestial biosphere, such as the El Niño-Southern Oscillation (ENSO), the Arctic Oscillation, and the North Atlantic Oscillation (e.g. Rödenbeck et al., 2008; Tohjima et al., 2015; Eddebbar et al., 2017).

There might be some link between ENSO and the $CO_2$ growth rates, for example, there is a large $CO_2$ growth rate in 2016 when it was a strong El Nino year (Fig. 14).

### 4.3 Seasonal cycles

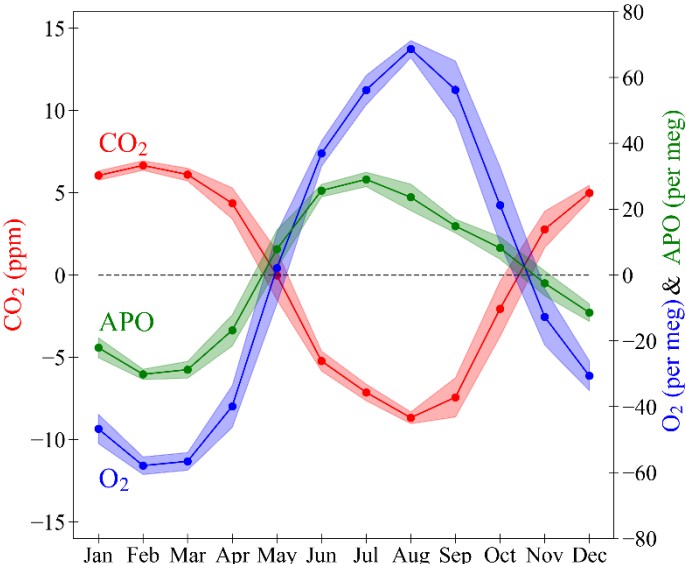

Figure 15: The seasonal cycles of $CO_2$ (red, left axis), $O_2$ (blue, right axis), and APO (green, right axis) at WAO.
Individual points are the monthly means of the seasonal component of the STL decomposition (see Sect. 2.3). The shaded bands are the average of the ±1σ standard deviation of the monthly averages of the baseline data. The y-axes are scaled to be visually comparable on a mole per mole basis.

The seasonal component of the WAO time series is obtained using the STL decomposition (see Sect. 2.3). The $CO_2$ mole
fraction is higher during the northern hemisphere winter and lower during the summer due mostly to the terrestrial biosphere (Fig. 15). Vegetation takes in $CO_2$ in the spring and summer in order to grow and releases $CO_2$ in the autumn and winter as it decomposes (Heimann et al., 1989). There is also an influence from increased temperature and sunlight during the summer leading to increased photosynthesis (Heimann et al., 1989). The exchange of $CO_2$ between the atmosphere and the oceans



takes ~1 year (Broecker and Peng, 1974) and so there is no discernible marine influence on the $CO_2$ seasonal cycle, and $O_2$
fluxes from different ocean processes are reinforcing on seasonal timescales, whereas for $CO_2$ these counteract (Keeling and
Manning, 2014).

The $O_2$ mole fraction is lower during the winter and higher during the summer. $O_2$ is anti-correlated with $CO_2$ as they are
involved in the same reactions, increased photosynthesis during the spring and summer releases $O_2$ and decomposition
during the autumn and winter takes in $O_2$ (Keeling and Shertz, 1992). The air-sea gas exchange of $O_2$ is much quicker than
for $CO_2$ and takes ~1 month (Broecker and Peng, 1974). During the summer $O_2$ is outgassed from the oceans and this
enhances the $O_2$ seasonal cycle and makes it larger than the $CO_2$ seasonal cycle (Keeling and Shertz, 1992).

APO is lower during the winter and higher during the summer. The seasonal cycle of APO is correlated with the seasonal
cycle of $O_2$ and anti-correlated with $CO_2$. APO is conservative with respect to terrestrial biosphere processes. The seasonal
cycle of APO is mainly driven by ocean ventilation changes, marine productivity cycling and the changing solubility of $O_2$ in
the oceans as temperature changes (Keeling et al., 1998). There is also a smaller effect on $CO_2$, $O_2$ and APO from the
seasonal cycle of fossil fuel emissions as more fossil fuel combustion takes place during the winter for heating and lighting
(Steinbach et al., 2011). There is also a seasonal cycle in the height of the atmospheric boundary layer, called the seasonal
rectifier effect, that influences all three species (Stephens et al., 2000). The APO seasonal cycle is about half the size of the
$O_2$ cycle (68/134= 0.5) indicating that about half of the $O_2$ seasonal cycle at WAO is due to terrestrial biosphere processes
and about half is due to ocean processes and fossil fuel combustion.

The $CO_2$ maximum/$O_2$ minimum/APO minimum occur between January and March each year (Table 6). The $CO_2$
minimum/$O_2$ maximum occur in August or September each year (Table 6). The APO maximum is more variable and tends to
occur earlier in the year between June and August (Table 6). The peak-to-peak amplitude of the seasonal cycle for $CO_2$ is 16
ppm (15 ppm to 17 ppm), for $O_2$ it is 134 per meg (128 per meg to 139 per meg), and for APO it is 68 per meg (62 per meg
to 73 per meg) (Fig. 16). A zero crossing day is the date at which the detrended curve crosses the x-axis (Keeling et al.,
1996b). Spring zero crossings occur in April or May each year and the autumn zero crossings in October or November each
year (Table 7).

**Table 6: Dates of the $CO_2$, $O_2$ and APO maximum and minimum values between 2011 and 2021 at WAO based on the
seasonal component of the STL decomposition. 2010 is not included as it is an incomplete year.**

| Year | $CO_2$ | $O_2$ | APO | $CO_2$ | $O_2$ | APO |
|---|---|---|---|---|---|---|
| | Maximum | Minimum | Minimum | Minimum | Maximum | Maximum |
| 2011 | 7 Mar | 13 Mar | 20 Mar | 5 Sep | 23 Aug | 2 Aug |
| 2012 | 11 Mar | 12 Mar | 20 Mar | 18 Aug | 21 Aug | 25 Jun |
| 2013 | 23 Mar | 12 Mar | 12 Mar | 19 Aug | 22 Aug | 25 Jun |
| 2014 | 11 Feb | 17 Feb | 12 Mar | 18 Aug | 22 Aug | 25 Jun |





| 2015 | 11 Feb | 18 Feb | 28 Feb | 25 Aug | 7 Sep  | 11 Jul |
| 2016 | 21 Feb | 18 Feb | 2 Mar  | 24 Aug | 30 Aug | 10 Jul |
| 2017 | 20 Feb | 16 Feb | 2 Mar  | 24 Aug | 20 Aug | 16 Jul |
| 2018 | 20 Jan | 16 Feb | 12 Feb | 20 Aug | 19 Aug | 18 Jul |
| 2019 | 21 Feb | 19 Feb | 12 Feb | 20 Aug | 13 Aug | 14 Aug |
| 2020 | 22 Feb | 1 Mar  | 10 Feb | 20 Aug | 3 Aug  | 9 Aug  |
| 2021 | 22 Feb | 1 Mar  | 4 Mar  | 20 Aug | 4 Aug  | 9 Aug  |


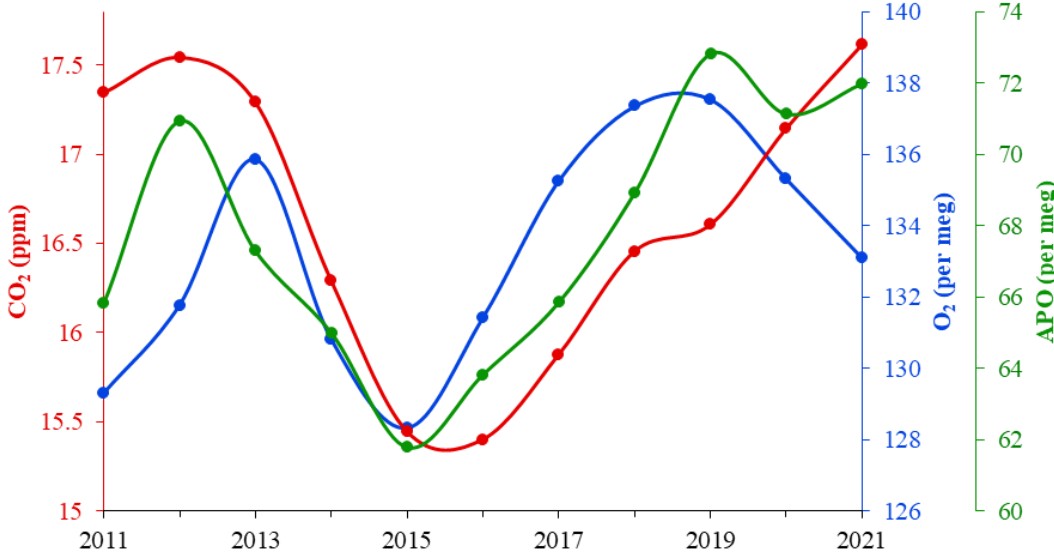

**Figure 16: Seasonal cycle peak-to-peak amplitudes for $CO_2$, $O_2$ and APO at WAO between 2011 and 2021 based on**
**the seasonal component of the STL decomposition. 2010 is not included as it is an incomplete year.**

**Table 7: Dates of the spring and autumn zero crossings for $CO_2$, $O_2$ and APO between 2011 and 2021 at WAO based**
**on the seasonal component of the STL decomposition. 2010 is not included as it is an incomplete year.**

| Year | $CO_2$ | $O_2$ | APO | $CO_2$ | $O_2$ | APO |
|------|--------|-------|-----|--------|-------|-----|
|      | Spring zero crossing | | | Autumn zero crossing | | |
| 2011 | 19 May | 13 May | 19 May | 27 Oct | 3 Nov  | 6 Nov  |
| 2012 | 24 May | 12 May | 26 Apr | 28 Oct | 2 Nov  | 4 Nov  |
| 2013 | 24 May | 12 May | 9 May  | 30 Oct | 1 Nov  | 2 Nov  |
| 2014 | 25 May | 11 May | 8 May  | 30 Oct | 30 Oct | 29 Oct |
| 2015 | 27 May | 11 May | 8 May  | 26 Oct | 29 Oct | 24 Oct |
| 2016 | 14 May | 13 May | 7 May  | 24 Oct | 31 Oct | 4 Nov  |
| 2017 | 14 May | 15 May | 5 May  | 26 Oct | 2 Nov  | 6 Nov  |
| 2018 | 13 May | 15 May | 6 May  | 27 Oct | 2 Nov  | 12 Nov |

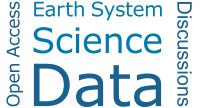

| | | | | | | |
|---|---|---|---|---|---|---|
| **2019** | 14 May | 15 May | 12 May | 25 Oct | 2 Nov | 14 Nov |
| **2020** | 14 May | 14 May | 11 May | 23 Oct | 3 Nov | 18 Nov |
| **2021** | 15 May | 15 May | 11 May | 29 Oct | 5 Nov | 20 Nov |

## 4.4 Seasonal variability of diurnal cycles

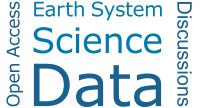


**Figure 17: The diurnal cycles of CO₂ (red, left axis), O₂ (blue, right axis), and APO (green, right axis) at WAO for each season: spring (MAM), summer (JJA), autumn (SON), and winter (DJF). Individual points are calculated using the 2-minute data from May 2010 to December 2021, with the timeVariation function in the OpenAir package in R (Carslaw and Ropkins, 2012). The shaded bands are the average of the ±1σ standard deviation of the hourly averages**
**of the data. The y-axes are scaled to be visually comparable on a mole per mole basis.**

In general, $CO_2$ mole fractions are lower during the day and higher at night (Fig. 17). $O_2$ mole fractions are anti-correlated with $CO_2$ and are higher during the day and lower at night. APO variations do not exhibit a diurnal cycle at WAO. The



diurnal cycles in $CO_2$ and $O_2$, like the seasonal cycles, are influenced by changes in the terrestrial biosphere processes of photosynthesis and respiration driven by changes in sunlight and temperature and are influenced by changes in the

atmospheric boundary layer height (Stephens et al., 2000; Kozlova et al., 2008). The atmospheric boundary layer is higher during the day and lower during the night, which causes a dilution effect, as molecules during the day are spread out over a larger volume (Denning et al., 1996; Larson et al., 2008). This amplifies the diurnal cycles in $O_2$ and $CO_2$ mole fractions and is referred to as the diurnal rectifier effect. Temperature variations in the oceans that affect gas solubility are not observable on diurnal timescales due to the higher heat capacity of water that can buffer temperature changes.

APO does not have a diurnal cycle because it is not influenced by the terrestrial biosphere; atmospheric boundary layer effects tend to cancel for APO on diurnal scales, since APO is the sum of two strongly anti-correlated species; and WAO is too remote for APO at the site to be influenced by the diurnal cycle in fossil fuel sources.

**Table 8: Hour in which the maximum and minimum mole fractions of $CO_2$ and $O_2$ occur for each season and the amplitude of the diurnal cycles. The $CO_2$ maximum and the $O_2$ minimum occur in the early morning and are**
**highlighted in orange. The $CO_2$ minimum and the $O_2$ maximum occur in the afternoon and are highlighted in grey. All times are UTC.**

| Species | Season | Maximum hour | Minimum hour | Amplitude (ppm or per meg) |
|---|---|---|---|---|
| CO₂ | Spring | 4am | 5pm | 7.3 |
| | Summer | 4am | 5pm | 12.6 |
| | Autumn | 6am | 2pm | 7.6 |
| | Winter | 6am | 2pm | 3.4 |
| O₂ | Spring | 3pm | 5am | 42 |
| | Summer | 5pm | 4am | 60 |
| | Autumn | 1pm | 6am | 43 |
| | Winter | 2pm | 6am | 19 |

There are differences in the diurnal cycles in different seasons. The average amplitude of the diurnal cycles of $CO_2$ and $O_2$ are largest in the summer ($CO_2$: 12.6 ppm, $O_2$: 60 per meg) and smallest in the winter ($CO_2$: 3.4 ppm, $O_2$: 19 per meg) (Table

8). This is due to the combined effect of the diurnal cycle and the seasonal cycle. In the summer, there are higher temperatures, more sunlight and longer days leading to more terrestrial biosphere productivity and larger changes in atmospheric boundary layer height as the differences in temperature between night and day are larger (Bakwin et al., 1995).

In the spring terrestrial productivity is starting to increase and in the autumn it is starting to decrease as plants die, lose their leaves are become less active. In the winter there are lower temperatures, less sunlight, and shorter days, leading to inactivity

of the terrestrial biosphere. Also, changes in atmospheric boundary layer height are smaller in the winter as the differences in





temperature between the night and day are smaller so the diurnal rectifier effect is less pronounced (Bakwin et al., 1995). So, the amplitudes of the $CO_2$ and $O_2$ diurnal cycles are smallest in the winter.

The timing of the maxima and minima also varies between seasons. There are more hours of daylight in the summer than in the winter so the $CO_2$ maximum/$O_2$ minimum occurs at 4am in the summer but 6am in the winter, as during the summer

photosynthesis starts earlier in the day. The $CO_2$ minimum/$O_2$ maximum occurs later during the summer at about 5pm and earlier in winter at about 2pm. As sunlight starts decreasing sooner in the winter so $CO_2$ increases/$O_2$ decreases earlier on as respiration takes over. The summer has a broad peak/trough in $O_2$/$CO_2$ during the day from about 6am to about 10pm (Fig. 17). Whereas in winter daytime peak/trough in $O_2$/$CO_2$ is shorter from about 10am to 8pm. Winter $CO_2$ and $O_2$ are mostly flat during the night from about 9pm to 9am as there is very little respiration taking place.

**5 Conclusions**

We have presented a 12-year time series of continuous measurements of $CO_2$, $O_2$ and APO at Weybourne Atmospheric Observatory (WAO) in the United Kingdom. The results from the GOLLUM, Cucumbers and WMO Round Robin intercomparison programmes show that WAO is sometimes but not always within the $O_2$ and $CO_2$ WMO compatibility goals established by the high-precision greenhouse gas measurement community. Measurements of Target Tanks indicate that the

2-minute $O_2$ repeatability is ±3.0 ± 4.6 per meg, averaged over the 12 years, which is within the extended repeatability goal of ±5 per meg. Target Tanks $CO_2$ repeatability is ±0.005 ± 0.023 ppm which is an order of magnitude below the repeatability goal (±0.05 ppm). Measurements of Zero Tanks indicate that the $O_2$ repeatability is ±3.1 ± 5.3 per meg and the $CO_2$ repeatability is ±0.005 ± 0.019 ppm which is similar to the $O_2$ repeatability indicated by the Target Tank measurements. We also show that the calibration coefficients vary over time and the $R^2$ of the $O_2$ calibration fit is >0.995, 92% of the time. The

Working Tanks show that $O_2$ mole fractions decrease in a cylinder as the pressure decreases. The ±1σ standard deviations of the air measurements for $CO_2$ are on average ±0.08 ± 0.10 ppm, based on the hourly averages, and are very stable for the whole time series, while the ±1σ standard deviations for the $O_2$ air measurements are more variable. Measurements are made every 2 minutes and between May 2010 and December 2021 there are 1,930,178 $CO_2$ measurements and 1,565,908 $O_2$ measurements. There are gaps in the dataset caused by regular calibrations, routine maintenance and technical issues, and

there are 6 gaps over the 12-year period that last longer than a month. Multiple technical issues in 2014 and 2015 decreased the precision of the data. There are no periods when the Target Tanks, Zero Tanks and calibration coefficients are all drifting at the same time indicating that the calibration scale at WAO is likely stable. The time series shows average long-term trends of 2.40 ppm yr$^{-1}$ (2.38 to 2.42) for $CO_2$, -24.0 per meg yr$^{-1}$ (-24.3 to -23.8) for $O_2$ and -11.4 per meg yr$^{-1}$ (-11.7 to -11.3) for APO. The average seasonal cycle peak-to-peak amplitude for $CO_2$ is 16 ppm (15 to 17), for $O_2$ it is 134 per meg (128 to

139), and for APO it is 68 per meg (62 to 73). The average amplitude of the diurnal cycle in summer is 12.6 ppm for $CO_2$ and 60 per meg for $O_2$ and the amplitude of the diurnal cycle in winter is 3.4 ppm for $CO_2$ and 19 per meg for $O_2$.



**Data availability**

The accompanying database comprises one csv file. The file contains information on the $CO_2$, $O_2$, and APO data (values and associated uncertainties) from Weybourne Atmospheric Observatory.

The file is published by the ICOS Carbon Portal, and is available at https://doi.org/10.18160/Z0GF-MCWH (Adcock et al., 2023).

Other data presented in this paper are available upon request.

**Author contribution**

ACM, EAK, AJM and AJE set up the original measurement system at the Weybourne Atmospheric Observatory. KEA, PAP,
ACM, GLF, LSF, TB, PAW, EAK, MH, AJE and AJM have all been involved in keeping the measurement system running, performing upgrades and improvements, assessing data quality, and carrying out data curation, at various points in time. ACM conceived the design of the system and bespoke software. AJE created and maintains the bespoke software used to run the measurement system. GLF manages the Weybourne Atmospheric Observatory. KEA wrote the original draft and KEA, PAP and ACM were involved in reviewing and editing the manuscript.

**Competing interests**

The authors declare that they have no conflict of interest.

**Acknowledgements**

The automated in situ atmospheric $O_2$ and $CO_2$ measurement system at WAO was built and maintained with assistance from Nick Griffin, Dave Blomfield, Gareth Flowerdew, Stuart Rix, and Nicholas Garrard (all staff at the University of East
Anglia). We are grateful to Michael Patecki (formerly at UEA), who played a key role in the original setup of the measurement system at WAO.

**Financial Support**

Atmospheric $O_2$ and $CO_2$ measurements at WAO were funded by the U.K. Natural Environment Research Council (NERC) grants NE/C002504/1, NE/F005733/1, NE/I013342/1, NE/I02934X/1, QUEST010005, NE/N016238/1, NE/S004521/1, and
NE/R011532/1, and by the EU FP6 Integrated Project CarboOcean (grant agreement no. 511176 GOCE). The WAO atmospheric $O_2$ and $CO_2$ measurements have also been supported by the National Centre for Atmospheric Science (NCAS). P. Wilson was supported by a NERC PhD studentship (NE/F005733/1) from 2008 to 2012. P. Pickers was supported by a



NERC PhD studentship (NE/K500896/1) from 2012 to 2016. T. Barningham was supported by a NERC PhD studentship (NE/L50158X/1) from 2014 to 2018. L. Fleming was supported by a NERC PhD studentship (NE/L002582/1) from 2018 to 2022. K. Adcock and P. Pickers received funding from the NERC project DARE-UK (NE/S004521/1). P. Pickers and A. Manning received support from the CHE project, funded by the European Union's Horizon 2020 Research and Innovation Programme under grant agreement no. 776186. K. Adcock, P. Pickers and A. Manning received funding from the Process Attribution of Regional Emissions project, funded by European Union's Horizon 2023 Research and Innovation Programme under grant agreement no. 101081430.

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
