# Peer review of "years of continuous atmospheric O2, CO2 and APO data from Weybourne Atmospheric Observatory in the United Kingdom"

_Earth System Science Data, 2023_

## Referee Comment (RC2)

**Review's comments**

**Manuscript Number:** essd-2023-129

**Title:** 12 years of continuous atmospheric $O_2$, $CO_2$ and APO data from Weybourne
Atmospheric Observatory in the United Kingdom

**Authors:** K. E. Adcock, P. A. Pickers, A. C. Manning, G. L. Forster, L. S. Fleming, T.
Barningham, P. A. Wilson, E. A. Kozlova, M. Hewitt, A. J. Etchells, and A. J. Macdonald

**General comments:**

The authors of this study present high-quality record of the atmospheric $CO_2$, $O_2$, and APO data
observed at Weybourne Atmospheric Observatory (WAO) in UK for decadal period between
May 2010 and December 2021. They carefully assess the stability of $CO_2$ and $O_2$ scales and the
repeatability and compatibility based on the measurements of variety of cylinders including
intercomparison round robin cylinders, Target Tanks, Zero Tanks, Working Secondary
Standards and so on. These results reveal that the data at WAO have high quality and
significantly reliable. They also investigate the characteristic features of the trend, seasonal
cycles, and diurnal variations of $CO_2$, $O_2$, and APO. The data at WAO would contribute to
various studies including the global carbon cycle, air-sea gas exchanges and so on. I found that
the paper is well written and contains material that should be published in Earth System Science
Data. I highly recommend the manuscript to be published with the minor corrections as
suggested below.

**Specific comments:**

Page 2, line 51: The authors described that a standard with a known $O_2/N_2$ ratio is used to report
the change in atmospheric $O_2/N_2$ ratio. Is it possible to show the exact number of the $O_2/N_2$ ratio
of the standard scale of this study?

Page 2, line 52: I think the sentence "$O_2$ and $N_2$ mole fractions are affected by changes in trace
gases" is a little misleading. The major atmospheric components like $O_2$ and $N_2$ are affected by
the change in the total amount of the air caused by changes in any atmospheric components,
which is called as a dilution effect. Therefore, $O_2$ mole fraction is affected not only by trace
gases, such as $CO_2$, but also $O_2$ itself. The dilution effect is, however, negligible for the trace

gases. Therefore, the direct comparison between $O_2$ and $CO_2$ concentrations is rather confusing when they are expressed as mole fractions.

Page 2, line 56-57: As far as I know, a mass spectrometric method, which is adopted by many laboratories, directly measure the $O_2/N_2$ ratio.

Page 2, line 58-59: The authors describe that $O_2$ variations are refer to as $O_2$ mole fraction changes rather than $\delta(O_2/N_2)$ ratio changes in this manuscript. But $\delta(O_2/N_2)$ ratios are used in the most of this manuscript.

Page 5, Figure 2: I think it would be better to add an aspirator and a differential pressure transducer in the legend.

Page 6, line 147-150: I'm curious about how to balance the pressures and flow rates between the sample air and WT air streams. In the manuscript, the authors described that the balance is manually achieved by adjusting the two needle valves. Is it possible to keep the balance for long period? In the Figure 2, the differential pressure transducer and the solenoid vale are connected to the "MKS" differential pressure gauge via green lines. Does it mean that the solenoid valve is automatically controlled to achieve the balance of the pressures between the sample air and WT air streams?

Page 6, line 149-150: Is "the two manual needles valves" a typo?

Page 6, line 152: Does "A solenoid valve" correspond to "4-way switching valve" in Figure 2? Are those same things?

Page 6: I think it would be better to clarify the flow rates of the sample air and WT air in this section of "Analytical set up". I know the flow rate (about 100 ml/min) is mentioned in in line 599, but it would be better to mention it here too.

Page 8, line 189-190: Don't the authors use the interpolated calibration coefficients from the bracketing calibrations?

Page 13, Figure 3: The shade of $\pm 10$ per meg range is unclear.

Page 14, line 340 (Figure 4 caption): "Target Tank (TT) measurements of $CO_2$ (top panel) and $O_2$ (bottom panel) at …"

Page 15, line 351-352: "… with slopes (in ppm year$^{-1}$ and per meg per year$^{-1}$ for $CO_2$ and $O_2$, respectively) …" "…each TT, for $CO_2$ (top panel) and $O_2$ (bottom panel) …"

Page 27, line 618-619: "Manning, 2001" is not listed in References.

Page 33, line 725-726: It would be better to clarify what the ranges in the parentheses mean. Are they 95% confidence intervals?

Page 35, line 767-768: I think that the effect derived from seasonal and/or diurnal covariance between surface fluxes and atmospheric transport including PBL dynamics is termed as rectification effect. The seasonal cycle of PBL height itself isn't termed as the rectification effect.

Page 46, line 1036-1037: "Stephens, B. B., …, 2000" has been already listed in line 1033-1035.

---

## Author Comment (AC1)

**Reviewer one comments**

Adcock et al present a 12 year time series of O2/N2, CO2, and the tracer APO from a coastal background site in the UK. This is a unique, high quality dataset highly worthy of publication in ESSD. I recommend publication after some revisions. My main issue with this paper is that it is overly long without being detailed enough. At least half of the paper is devoted to an analysis of the dataset, which per the ESSD guidelines is not supposed to be included. This seems to come at the expense of detailed information on changes in the measurement system and technical issues, which are not fully described and could be of interest to anyone trying to actually use the data. My suggestion is to generate a complete list of calibration, target, and zero tank changes with IDs and assigned values (where applicable), and a complete change log or README type file where the major alterations to the measurement system are fully described, with exact dates. This shouldn't be too difficult to generate given the figures presented (i.e. the authors must have this information in hand to create the figures). There is a long paragraph of significant changes with approximate dates which would already form the basis of such a table. I am asking for this because in 10 or 20 years, someone may wish to analyze the time series but not have enough information as presented to understand whether a given feature is a real signal or an artifact. The paper also contains a lot of verbalization of data which is already in a table. It's a very long paper, and cutting this redundant text will make for less text to sift through.

We thank the reviewer for their positive comments and thorough review of our manuscript which has helped us to improve the text and figures. We will address the general points raised below.

Regarding the comment about the amount of analysis of the dataset in our manuscript. We think that there is a difference between describing the key features of the dataset, and actually analysing or interpreting the dataset. For example, we think it's acceptable to talk about the average long-term trends and seasonal cycles and to describe in general terms what is causing these. Other ESSD articles have included similar descriptions of key features e.g. Nguyen et al., 2022; Friedlingstein et al., 2022, and so we do not consider our manuscript to be out of the ESSD scope.

Nguyen, L. N. T., et al.: Two decades of flask observations of atmospheric δ(O2/N2), CO2 and APO at stations Lutjewad (the Netherlands) and Mace Head (Ireland), and 3 years from Halley station (Antarctica), 14, 991–1014, https://doi.org/10.5194/essd-14-991-2022, 2022.

Friedlingstein, P., et al.: Global Carbon Budget 2022, Earth Syst. Sci. Data, 14, 4811–4900, https://doi.org/10.5194/essd-14-4811-2022, 2022.

Therefore, we don't think we need to remove all of Section 4; however, we have decided to shorten and reword some of the text, and to exclude some figures and tables that present more than the basic features of the dataset. Consequently, we have removed Figure 14 (interannual variability of long-term trends), Table 6 (seasonal maximum & minimums), Table 7 (zero crossing dates), Figure 16 (interannual variability in seasonal amplitude) and Table 8 (diurnal cycles).

We respectfully disagree with the reviewer's comment that a table of all the cylinders and a change log should be added to the manuscript. This is not something that is typically expected for measurement-related papers. It is unlikely that we could document all these changes thoroughly in a paper in a way in which the reader would be able to understand the implications on the data quality. The data have undergone full quality control and we would not publish them if we were concerned that such changes to the measurement system had compromised the data quality. We encourage data users to contact us if they have questions about specific features of the dataset.

We agree with the reviewer's comment that there is some redundant verbalization of the tables and figures, so we have cut some of the text. Please see our replies below for more information on specific sections.

1. My last major comment is that the correction to the APO record is speculative at best. Forcing the WAO data to fit the CBA record is a creative approach, but the impact this has on the WAO data was not demonstrated. As I understand it it relies on the assumption that fractionation through a torn diaphragm was constant over shorter time scales, but not over several months. Is there any basis for making this determination? I understand the desire to salvage a significant chunk of data, and I think this is generally an OK approach, but the authors could have done more to convince me that it was reasonable. Could they perform the same baseline shift exercise on data which was not impacted at WAO and see how the residuals compare--do they have any structure? And what do the corrected vs uncorrected residuals look like? At the very least, this data should be better flagged in the data file--right now it is flagged as "2", which means "contact data provider". A separate flag for "corrected" or "baseline shifted" should be implemented.

We understand that the reviewer has some concerns about the adjustment applied to the 2018/2019 data. We hope that we can allay these concerns. Firstly, it is not uncommon to correct periods of data in this way, for example, Max Planck Institute for Biogeochemistry (MPI-BGC) used a similar method to adjust their $O_2$ flask data after they discovered a leak in a valco valve (Rödenbeck et al., 2023). Secondly, we already provided detail regarding the quality control checks we carried out to ensure our pump correction was reasonable, as shown in Section S1 of the supplement. We used the Cold Bay Alaska data to apply the adjustment, and then used the Shetland Islands data to check that the adjustment was reasonable.

Rödenbeck, C., et al., The suitability of atmospheric oxygen measurements to constrain Western European fossil-fuel $CO_2$ emissions and their trends, EGUsphere [preprint], https://doi.org/10.5194/egusphere-2023-767, 2023.

We have included a plot below of the data excluding baseline variability for the adjusted-leak data and for the 9 months after the leak, to show that the variation excluding the baseline is similar for both time periods.

[Figure]

[Figure]

We have also included a plot below showing the data excluding the baseline for the leak period, before the adjustment was applied and afterwards, to show that the variation excluding the baseline is similar in both cases.

[Figure]

Residuals before and after adjustment

We would prefer to keep the data flag as "contact data provider" as this is consistent with the set of standardised flags we are required to use when we upload these data to community databases and repositories. We strongly encourage all users of the data to contact us in any case, and we are happy to discuss the pump correction period with data users directly and in relation to their specific analysis if they have additional queries.

Minor comments:

2. General comment: The use of O2 mole fraction throughout is confusing. It is a term with a specific meaning, and I don't find it more convenient to alter its meaning for this particular paper.

While we appreciate our use of $O_2$ mole fraction may be confusing to those outside of the $O_2$ measurement community, there is a long and established precedent for using mole fraction to refer to atmospheric $O_2$ measurements in the existing literature, e.g.:

- Keeling, R. F.: Measuring correlations between atmospheric oxygen and carbon dioxide mole fractions: A preliminary study in urban air, 7, 153–176, https://doi.org/10.1007/BF00048044, 1988.
- Tohjima, Y., Machida, T., Watai, T., Akama, I., Amari, T., and Moriwaki, Y.: Preparation of gravimetric standards for measurements of atmospheric oxygen and reevaluation of atmospheric oxygen concentration, J. Geophys. Res. D Atmos., 110, 1–11, https://doi.org/10.1029/2004JD005595, 2005.

We would prefer not to deviate from the established practice of our community and wish to continue to use the term $O_2$ mole fraction in our manuscript. In some places, it would be incorrect and misleading to use the term $\delta(O_2/N_2)$ because our analyser does not directly measure $\delta(O_2/N_2)$ ratios, and because we convert our $O_2$ mole fraction measurements in units of "ppm equivalent" to per meg values. We would also prefer not to use the term 'concentration', because it is generally only accepted that this is used when communicating to non-scientific audiences, as mentioned in the latest WMO Global Atmosphere Watch report on GHG and related tracer measurement techniques (WMO report #255).

3. General comment: The figures are not color blind safe, please use different symbols in addition to the colors selected, or use a different pallette.

We agree with the reviewer and have amended figures 1, 3, 4, 7, 8, 9, 10, 14 and 15 to be more colourblind friendly, please see the amended version of the manuscript.

4. L16: APO is not a tracer for terrestrial biosphere processes, this would be better phrased as "insensitive to terrestrial biosphere fluxes".

We have decided to change the word "conservative" to "invariant", so we changed the phrase from:

a tracer that is conservative with respect to terrestrial biosphere processes

To:

a tracer that is invariant to terrestrial biosphere fluxes

5. L27-47: This is a fine introduction, but the paper is 47 pages long and its goal is simply to describe the data set. These two paragraphs could be consolidated into a sentence or two, pointing to some key references. I don't think it's necessary to list all of the ORs for different fuel types, for instance.

We agree with the reviewer and have shortened these two paragraphs and combined them into one paragraph.

6. L52: A good place to include the equation for O2/N2

We have added the equation for $O_2/N_2$, and also the reference:

As such, atmospheric $O_2$ mole fractions are typically reported as changes in the ratio of $O_2$ to $N_2$, relative to a reference $O_2/N_2$ ratio (Keeling and Shertz, 1992).

$$\delta(O_2/N_2) = \frac{(O_2/N_2)_{sample} - (O_2/N_2)_{reference}}{(O_2/N_2)_{reference}} \tag{1}$$

7. L58-59: I don't understand, per meg is used throughout the paper...?

Per meg is the unit we use for $O_2$, in the same way that ppm is the unit we use for $CO_2$, whereas "$O_2$ mole fraction" is the name of what we are actually measuring in "ppm equivalent" units, before we convert $O_2$ into per meg units. To make this clearer, we have shortened the sentence about this from:

For simplicity, in this paper we refer to $O_2$ variations as $O_2$ mole fraction changes rather than $\delta(O_2/N_2)$ ratio changes.

To:

For simplicity, in this paper we refer to $O_2$ variations as $O_2$ mole fraction changes.

8. L60-70: Suggest to cut this paragraph, it's not necessary to explain the data being presented.

We agree with this comment and have cut this paragraph.

9. L71: "Calculated term" is awkward, suggest "tracer" or "data-derived tracer"

We respectively disagree with the reviewer as APO is a calculated term and we don't think that "data-derived tracer" is less awkward than "calculated term". "Calculated term" is not incorrect, and we would rather not change it.

10. L75: this is not what I understand a "conservative tracer" to mean

We have changed "conservative with respect" to "invariant"

11. L125: How long is the inlet line? The mast height is given, but not the total distance from the inlet to the instruments.

    L126: Could you briefly describe the inlet? Dimensions, how and how much it is aspirated, etc. What is the flow rate in the sample line?

We have added in this information and the paragraph has been changed from:

There are two inlet lines (Synflex, type 1300 tubing, ¼" OD) and the measurement system alternates sampling air between these two inlet lines every two hours to diagnose for leaks, blockages, or other faults. Each inlet line includes an aspirated air inlet, to avoid fractionation of $O_2$ molecules relative to $N_2$ (Blaine et al., 2006) and a small diaphragm pump (KNF Neuberger Inc.; model PM27653-N86ATE) to draw air through the inlet tubing.

To:

There are two inlet lines, and the measurement system alternates sampling air between these two inlet lines every two hours to diagnose for leaks, blockages, or other faults. The inlet lines are approximately 14 meters long and are made of Synflex type 1300 tubing with an outer diameter of 1/4". Each inlet line includes an aspirated air inlet (Aspirated Radiation Shield Model No. 43502, Read Scientific Ltd.), whereby the inlet samples from a moving airstream and shields the entrance from solar radiation, to avoid fractionation of $O_2$ molecules relative to $N_2$ (Blaine et al., 2006) and a small diaphragm pump (KNF Neuberger Inc.; model PM27653-N86ATE) to draw air through the inlet tubing at 100 mL min$^{-1}$.

We also, added in a sentence about the flow rate later in this section:

The flow rate of the measurement system (both the sample and working reference sides) is set to 100 mL min$^{-1}$ using a mass flow controller (MFC, Fig. 2).

12. L298-314: This paragraph could be shortened or cut by putting the data in a table. It's not necessary to state after each result whether it is smaller or larger than 2 or 10, the reader can do this on their own. The statement that "...any systematic drift over time, indicating long-term stability of the WAO O2 and CO2 calibration scales" is quite misleading. Both the CCL scale and WAO scale could be drifting together, and the drift does not necessarily have to be systematic. Scales can drift over multiple time scales for many different reasons, which can appear to be scatter when sampled sparsely.

We agree with this comment and have added a table showing the intercomparison results for $CO_2$ and $O_2$. Additionally, we removed a few sentences from the intercomparison section, including the sentences comparing to 2 and 10 per meg and the sentence about systematic drift.

13. Fig 4: Isn't the target tank a measure of the repeatability, not the compatibility? If so, the shaded bands should be half the compatibility goal as pointed out on L263.

The TTs are used both to measure the repeatability and the compatibility. The compatibility is how much in agreement the measurements at WAO are to the UEA CRAM Laboratory over both long and short-term timeframes, where TT cylinders are usually analysed before being used at WAO. In Figure 4, the UEA CRAM Laboratory values are zero on the y-axes, so Figure 4 is showing the compatibility of the TTs, and therefore the shaded band is the right size. The repeatability is calculated using the mean of $\pm1\sigma$ standard deviations for every two consecutive 2-minute averages for each TT run. So, the repeatability can be thought of more as a measure of short-term reproducibility under the same conditions. The repeatability is shown in Table 2.

We are using the definitions of compatibility and repeatability that come from the 20th WMO/IAEA Meeting on Carbon Dioxide, Other Greenhouse Gases and Related Measurement Techniques (GGMT-2019), Global Atmosphere Watch (GAW) Report: compatibility is a measure of the persistent bias between measurement records; and repeatability is a measure of the closeness of the agreement between the results of successive measurements over a short period of time (Crotwell et al., 2020).

14. Fig 5: The drift in the cylinders is not linear (except for maybe the last one), so I question the usefulness of showing linear fits to them. If the tanks are desorbing/fractionation with pressure, one wouldn't expect it to drift linearly anyway, unless it was leaking badly. Finally, this figure doesn't show any information not already contained in Fig 4. I recommend cutting it, or at least consolidating with Fig 4 by adding the fits to the data.

We thank the reviewer for this comment. We think that combining Fig. 4 and Fig. 5 makes the figure look cluttered and would prefer to keep them separate. We think that Fig. 5 does show something different to Fig. 4, for example, in Fig. 5 you can clearly see from the trend lines, for $CO_2$ for TT02 and TT07 that there is a small offset between the UEA CRAM Laboratory and WAO of about 0.1 ppm, which is not as easy to see in Fig. 4 because of the scatter of the data points.

We are not implying that the trends in the tanks are linear but have used linear fits to approximate the drifts over time in order to assess how significant these are within the context of WMO compatibility goals.

[Figure]

15. L395-419: This section could also be cut down, given that all of this information is in the table.

We agree with this comment and have reduced the text of this section accordingly.

16. L448: These are per meg values, not mole fractions...maybe this is what is meant in L58? If so, it is confusing and looks like an error. "Value" would work fine here.

We agree that this is confusing and incorrect, as we should not have combined the term $O_2$ mole fraction with the unit of per meg. We have changed the text from:

Excluding ZT20, which had an $O_2$ mole fraction of 294 per meg, the ZT $O_2$ mole fraction ranged from -478 per meg to -1363 per meg.

To:

Excluding ZT20, which had an $O_2$ value of 294 per meg, the ZT $O_2$ values ranged from -478 per meg to -1363 per meg.

17. L474-479: There is a cumbersome emphasis here on the compatibility goal, usually with no added discussion. There is also a repeated pattern of verbally describing the values in a table within the text. I think all of this can be cut.

We have shortened and reworded this paragraph, from:

If we exclude the 3 ZTs that were vertical, the ZT repeatability is on average for $O_2$ ±3.1 ± 5.3 per meg, which is more than the ±1 per meg repeatability goal, and less than the extended repeatability goal of ±5 per meg, but the ±1σ standard deviation is higher. The $O_2$ repeatability based on the ZTs is similar to the $O_2$ repeatability based on the TTs (±3.0 ± 4.6 per meg, see Sect. 3.2), which increases our confidence, that this is the $O_2$ repeatability of the measurement system. The average $CO_2$ repeatability based on the ZTs is ±0.005 ± 0.019 ppm, which is again similar to the $CO_2$ repeatability based on the TTs, (±0.005 ± 0.023 ppm, see Sect. 3.2).

To:

> If we exclude the 3 ZTs that were vertical, the average $CO_2$ and $O_2$ repeatability based on the ZTs is $\pm 0.005 \pm 0.019$ ppm and $\pm 3.1 \pm 5.3$ per meg, respectively, which is similar to the $CO_2$ and $O_2$ repeatability based on the TTs ($\pm 0.005 \pm 0.023$ ppm and $\pm 3.0 \pm 4.6$ per meg, see Sect. 3.2). These results increase our confidence, that this is the repeatability of the measurement system.

18. L533: Calibration cylinders should be remeasured to account for cylinder drift. I think the authors should try to constrain how much this contributes to the uncertainty. Also, From Fig 8 it looks like the changes in the slope are small due to changing of calibration tanks, but it should be shown that this is a small effect. Have the authors indicated whether the calibration coefficients are interpolated between calibrations, or applied as step changes? I may have missed this.

We thank the reviewer for this comment. It is already stated in the text that the calibration coefficients are applied as step changes and not interpolated between calibrations, in lines 189-190:

> With the exception of the $CO_2$ c-term, the calibration coefficients are redetermined every 47 hours, and then these new values are used until the next calibration.

We agree with the reviewer that ideally the calibration cylinders would be remeasured to account for cylinder drift, but for practical reasons it has not been possible to do this for Weybourne yet, though we still hope to measure some of our TTs in the near future. However, we have no reason to believe that the WAO scale has drifted relative to the UEA CRAM Laboratory or to the Scripps Institution of Oceanography scale, as shown by the intercomparison programme results.

In lines 504-506, we state:

> A drift in the calibration scale may be caused by internal drift of the mole fractions in the calibration cylinders. This can only be determined by reanalysing the calibration cylinders after they have finished being measured at WAO. However, due to practical constraints this is not done on the calibration cylinders used at WAO.

We have added a couple of sentences to the end of this paragraph for clarification:

> While, we cannot know if the calibration cylinders have drifted over time, if they had drifted we would expect to see a noticeable step change in the air measurements when the calibration cylinders are changed, and/or visible drift in the TT measurements (depending on the rate of the drift). In addition, we would expect to see evidence of long-term drift in our intercomparison programme results.

19. L673-688: The lengthy text on technical issues should be at least separated from the text on missing APO data. I think this information needs to be formatted into a table or change log of some kind, with exact dates, spanning the whole dataset. As it is only approximate dates are given and the user would have to guess if subtle features in the dataset might be artifacts pertaining to such changes. There is also not enough detail--for instance, "pneumatic valves" and "solenoid valves". Could the authors reference the plumbing diagram directly in a way which is unambiguous?

Please, see our reply above that discusses this. We do not believe it is necessary to include a change log, as this is not something that is typically expected for a measurement related paper, it

would be unlikely that we could include enough detail to ensure that readers would be able to understand what the potential impacts on the data could be. The data have been rigorously quality controlled to ensure the data have not been unduly impacted by technical changes to the system and any data that have been impacted have been removed. The technical issues that have led to large gaps in the data, are described in the text because it is necessary to explain to the reader why those large gaps in the data exist.

We have moved the missing APO data text and the technical issues text into separate paragraphs and have added more detail about the pneumatic valves and solenoid valves and referenced the plumbing diagram.

20. L687: "was not so it had to be removed from the dataset." -- missing word or typo?

We thank the reviewer for spotting this error and we have reworded this sentence.

21. L703-713: This is analysis of the dataset, per the "Aims and Scope" of ESSD: "Any interpretation of data is outside the scope of regular articles."

As mentioned above, we agree with the reviewer that some of Section 4 is quite scientific for an ESSD paper. This paragraph has been shortened and reworded and combined with the first paragraph in Section 4.2. Thus, we are only describing the key features of the dataset, rather than analysing or interpreting the underlying science.

We have changed this paragraph from:

> From Fig. 13 we can see many interesting features of this dataset. Firstly, there are long-term trends in the measurements with increasing mole fractions of $CO_2$ and decreasing mole fractions of $O_2$ and APO, predominantly due to the burning of fossil fuels and land-use changes which release $CO_2$ and consume $O_2$ (Sect. 4.2). These trends are buffered by an increasing land carbon sink, and the ocean carbon sink is also buffering the $CO_2$ trend but not the $O_2$ trend (Keeling et al., 1996a). Secondly, there are seasonal cycles: $CO_2$ has higher mole fractions during the winter and lower mole fractions during the summer, while the seasonal cycles of $O_2$ and APO are reversed, with lower mole fractions during the winter and higher mole fractions during the summer (Sect. 4.3). These seasonal cycles are caused by seasonal changes in terrestrial biosphere processes, ocean processes and fossil fuel emissions. Thirdly, it can also be seen that there is a lot of synoptic variability in the species measured at WAO, due to the influence of more local emissions within the atmospheric footprint. The spikes are less common in the APO time series therefore indicating that most of this variability is mostly from terrestrial biosphere processes, and not from the ocean.

To:

> From Fig. 13 we can see there are long-term trends in all three species. On average, atmospheric $CO_2$ at WAO increased by 2.40 ppm yr$^{-1}$ (2.38 to 2.42; 95% confidence intervals), atmospheric $O_2$ decreased by 24.0 per meg yr$^{-1}$ (24.3 to 23.8) and APO decreased by 11.4 per meg yr$^{-1}$ (11.7 to 11.3). The long-term trends were calculated using the trend from the seasonal decomposition with STL (see Sect. 2.3) and the TheilSen function in the openair package in R (Carslaw and Ropkins, 2012; Carslaw 2019). These long-term trends are predominantly due to the burning of fossil fuels and land-use changes which release $CO_2$ and consume $O_2$. Atmospheric $O_2$ is decreasing more quickly than $CO_2$ is increasing because the $CO_2$ increase is buffered by an increasing land carbon sink and ocean carbon sink, whereas the $O_2$ decrease is only buffered by an increasing land oxygen source and a small ocean oxygen source from $O_2$ outgassing (Keeling et al., 1996a).

22. Section 4.2: Same comment as previous, also noting the ESSD "Aims and Scope" statement: "Any comparison to other methods is beyond the scope of regular articles".

As mentioned above, we agree with the reviewer that some of this text is quite scientific for an ESSD paper. As such, we have removed Figure 14 (interannual variability of long-term trends), as well as the second paragraph in Section 4.2, which discusses the interannual variability of the long-term trends and their potential link to ENSO. We have also reworded the first paragraph and combined it with the last paragraph in Section 4.1, so this Section 4.2 no longer exists.

23. L754: "there is no discernible marine influence on the CO2 seasonal cycle" -- This is not correct, there is a significant component to the seasonal cycle of atmospheric CO2 from air-sea fluxes. If the authors mean no discernible contribution at WAO, this would be surprising given its location, and needs to be shown.

We respectfully disagree with the reviewer regarding this comment. Perhaps the reviewer is thinking of $CO_2$ fluxes from the ocean, which do have seasonality, whereas we are referring to the variability of $CO_2$ mole fractions in the atmosphere. The diagram below visualises the point we are making in the text.

[Figure]

We have made minor amendments to the text with additional references. We have changed this sentence from:

> The exchange of $CO_2$ between the atmosphere and the oceans takes ~1 year (Broecker and Peng, 1974) and so there is **no discernible marine influence on the $CO_2$ seasonal cycle**, and $O_2$ fluxes from different ocean processes are reinforcing on seasonal timescales, whereas for $CO_2$ these counteract (Keeling and Manning, 2014).

To:

> The exchange of $CO_2$ between the atmosphere and the oceans takes ~1 year (Broecker and Peng, 1974), and $O_2$ fluxes from different ocean processes are reinforcing on seasonal timescales whereas for $CO_2$ these counteract (Keeling and Manning, 2014). The combined effect of the $CO_2$ lag and counteracting $CO_2$ processes is that there is **a minimal marine influence on the atmospheric $CO_2$ mole fraction seasonal cycle (Heimann et al., 1989; Randerson et al., 1997)**.

24. L805: APO has a diurnal cycle in many locations, better to specify "APO at WAO" to avoid confusion. One would in fact expect a strong diurnal cycle in APO at a coastal site like WAO

due to land/sea breezes. It is also not correct that the boundary layer effects cancel. The changing boundary layer height dilutes or concentrates a flux signal from the surface with background air from higher up in the troposphere. If you had constant outgassing of APO from the ocean surface over 24 hours, for instance, you would still see a diurnal cycle from this effect.

We have changed this sentence from "APO does not…" to "APO at WAO does not…". However, we respectfully disagree with the reviewer regarding the comment about expecting a strong diurnal cycle in APO. As shown in Fig. 15 there is not a strong diurnal cycle in APO at WAO. Our findings are consistent with other studies (e.g. Kozlova et al., 2008; Wilson, 2012; Pickers, 2016) which show that APO does not exhibit a strong diurnal cycle in the same way that $CO_2$ and $O_2$ do, because APO is invariant to terrestrial processes, and because ocean fluxes (even for $O_2$) are not instantaneously realised in the atmosphere in the same way terrestrial fluxes are due to air-sea gas exchange processes. APO fluxes from the ocean surface are relatively very small on synoptic scales compared to terrestrial $O_2$ and $CO_2$ fluxes, and boundary layer height variations over the ocean are also much smaller compared to those over land. Our research group has many years' experience making high-precision $O_2$ and $CO_2$ measurements over the ocean (e.g. Pickers et al. 2017) and we do not find any evidence of APO diurnal cycles in these data.

---

## Author Comment (AC2)

**Reviewer two comments**

**Manuscript Number:** essd-2023-129

**Title:** 12 years of continuous atmospheric $O_2$, $CO_2$ and APO data from Weybourne Atmospheric Observatory in the United Kingdom

**Authors:** K. E. Adcock, P. A. Pickers, A. C. Manning, G. L. Forster, L. S. Fleming, T. Barningham, P. A. Wilson, E. A. Kozlova, M. Hewitt, A. J. Etchells, and A. J. Macdonald

**General comments:**

The authors of this study present high-quality record of the atmospheric $CO_2$, $O_2$, and APO data observed at Weybourne Atmospheric Observatory (WAO) in UK for decadal period between May 2010 and December 2021. They carefully assess the stability of $CO_2$ and $O_2$ scales and the repeatability and compatibility based on the measurements of variety of cylinders including intercomparison round robin cylinders, Target Tanks, Zero Tanks, Working Secondary Standards and so on. These results reveal that the data at WAO have high quality and significantly reliable. They also investigate the characteristic features of the trend, seasonal cycles, and diurnal variations of $CO_2$, $O_2$, and APO. The data at WAO would contribute to various studies including the global carbon cycle, air-sea gas exchanges and so on. I found that the paper is well written and contains material that should be published in Earth System Science Data. I highly recommend the manuscript to be published with the minor corrections as suggested below.

We thank the reviewer for their positive review and assessment of our manuscript which has helped us to improve the text and figures. We will address the points raised below.

**Specific comments:**

1. Page 2, line 51: The authors described that a standard with a known $O_2/N_2$ ratio is used to report the change in atmospheric $O_2/N_2$ ratio. Is it possible to show the exact number of the $O_2/N_2$ ratio of the standard scale of this study?

   We realise that the way this is written is misleading, so we have changed this sentence from:

   As such, atmospheric $O_2$ mole fractions are typically reported as changes in the ratio of $O_2$ to $N_2$, relative to a standard with a known $O_2/N_2$ ratio.

   To:

   As such, atmospheric $O_2$ mole fractions are typically reported as changes in the ratio of $O_2$ to $N_2$, relative to a reference $O_2/N_2$ ratio. This study uses a $O_2/N_2$ reference derived from a suite of compressed air reference gases stored in high-pressure tanks and maintained but the Scripps Institution of Oceanography, U.S.A. (SIO; Sect. 2.2; Keeling et al., 2007).

2. Page 2, line 52: I think the sentence "$O_2$ and $N_2$ mole fractions are affected by changes in trace gases" is a little misleading. The major atmospheric components like $O_2$ and $N_2$ are affected by the change in the total amount of the air caused by changes in any atmospheric components, which is called as a dilution effect. Therefore, $O_2$ mole fraction is affected not only by trace gases, such as $CO_2$, but also $O_2$ itself. The dilution effect is, however, negligible for the trace gases. Therefore, the direct comparison between $O_2$ and $CO_2$ concentrations is rather confusing when they are expressed as mole fractions.

We agree with the reviewer and have changed this sentence from:

> $O_2$ and $N_2$ mole fractions are affected by changes in trace gases, such as $CO_2$, since mole fractions are relative to the total amount and therefore changing the total number of molecules in the air will make it appear as if the amount of $O_2$ and $N_2$ are changing even when they are not. Reporting $O_2$ as the $O_2/N_2$ ratio circumvents this issue.

To:

> Since mole fractions are relative to the total amount of air, changing the total number of molecules, for example by changing the number of $CO_2$ molecules, will make it appear as if the amount of $O_2$ and $N_2$ are changing even when they are not. This dilution effect is problematic for $O_2$ and $N_2$ because they are not trace gases, however, reporting $O_2$ as the $O_2/N_2$ ratio circumvents this issue. The dilution effect also exists for trace gases but has a negligible effect.

3. Page 2, line 56-57: As far as I know, a mass spectrometric method, which is adopted by many laboratories, directly measure the $O_2/N_2$ ratio.

$CO_2$ still needs to be measured to do a correction even when using a mass spec. So, this sentence needs to be changed as the point about measuring $CO_2$ is not really dependent on whether the $O_2/N_2$ ratio is measured directly or not.

We have changed this sentence from:

> In practice most analytical techniques in use do not measure the $O_2/N_2$ ratio directly and therefore when measuring $O_2$, $CO_2$ must also be measured concurrently, and a correction applied to account for changes in $CO_2$.

To:

> When measuring $O_2$, $CO_2$ must also be measured concurrently, and a correction applied to account for changes in $CO_2$.

4. Page 2, line 58-59: The authors describe that $O_2$ variations are refer to as $O_2$ mole fraction changes rather than $\delta(O_2/N_2)$ ratio changes in this manuscript. But $\delta(O_2/N_2)$ ratios are used in the most of this manuscript.

It's common in the literature to refer to atmospheric $\delta(O_2/N_2)$ measurements as $O_2$ mole fractions to simplify the text and because some methods do not measure the $\delta(O_2/N_2)$ ratio directly.

A couple of examples of articles that refer to $O_2$ mole fractions are:

- Keeling, R. F.: Measuring correlations between atmospheric oxygen and carbon dioxide mole fractions: A preliminary study in urban air, 7, 153–176, https://doi.org/10.1007/BF00048044, 1988.
- Tohjima, Y., Machida, T., Watai, T., Akama, I., Amari, T., and Moriwaki, Y.: Preparation of gravimetric standards for measurements of atmospheric oxygen and reevaluation of atmospheric oxygen concentration, J. Geophys. Res. D Atmos., 110, 1–11, https://doi.org/10.1029/2004JD005595, 2005.

Per meg is the unit we use for $O_2$, in the same way that ppm is the unit we use for $CO_2$, whereas "$O_2$ mole fraction" is the name of what we are actually measuring in "ppm equivalent" units,

before we convert $O_2$ into per meg units. To make this clearer, the sentence about this has been shortened, from:

> For simplicity, in this paper we refer to $O_2$ variations as $O_2$ mole fraction changes rather than $\delta(O_2/N_2)$ ratio changes.

To:

> For simplicity, in this paper we refer to $O_2$ variations as $O_2$ mole fraction changes.

5. Page 5, Figure 2: I think it would be better to add an aspirator and a differential pressure transducer in the legend.

We agree with the reviewer and have added these two things to the legend in Figure 2.

6. Page 6, line 147-150: I'm curious about how to balance the pressures and flow rates between the sample air and WT air streams. In the manuscript, the authors described that the balance is manually achieved by adjusting the two needle valves. Is it possible to keep the balance for long period? In the Figure 2, the differential pressure transducer and the solenoid vale are connected to the "MKS" differential pressure gauge via green lines. Does it mean that the solenoid valve is automatically controlled to achieve the balance of the pressures between the sample air and WT air streams?

The two needle valves allow for fine control of the restriction on each side to ensure that the matched pressures do result in matched flows, however, we agree that the way this was written before gave the impression that the pressure balance is achieved manually, which is not correct. We have therefore changed this sentence from:

> This balance is achieved with a differential pressure transducer (MKS Instruments, model Baratron 223B, ±10 mbar full scale range) and the two manual needles (Brooks Instrument, model 8504) valves immediately downstream of the Ultramat (Fig. 2).

To:

> This balance is achieved with a differential pressure transducer (MKS Instruments, model Baratron 223B) which measures the pressure difference between the sample and WT air streams, and then adjusts the sample side pressure to match the pressure of the WT air using a fast-response solenoid valve (MKS Instruments Inc., 248A; Fig. 2).

7. Page 6, line 149-150: Is "the two manual needles valves" a typo?

We have removed this phrase. Please see the reply above.

8. Page 6, line 152: Does "A solenoid valve" correspond to "4-way switching valve" in Figure 2? Are those same things?

The 4-way switching valve is a type of solenoid valve, we have changed "A solenoid valve" to "A 4-way switching solenoid valve".

9. Page 6: I think it would be better to clarify the flow rates of the sample air and WT air in this section of "Analytical set up". I know the flow rate (about 100 ml/min) is mentioned in in line 599, but it would be better to mention it here too.

We agree with the reviewer and have added a sentence about the flow rate to this section.

The system flow rate is established on the WT side of the system, to 100 mL/min using a mass flow controller (MFC, Fig. 2).

10. Page 8, line 189-190: Don't the authors use the interpolated calibration coefficients from the bracketing calibrations?

As mentioned in line 189-190, we use the calibration coefficients from the most recent calibration:

With the exception of the $CO_2$ c-term, the calibration coefficients are redetermined every 47 hours, and then these new values are used until the next calibration.

11. Page 13, Figure 3: The shade of $\pm10$ per meg range is unclear.

We have made the shading into a darker grey to make it more visible.

12. Page 14, line 340 (Figure 4 caption): "Target Tank (TT) measurements of $CO_2$ (top panel) and $O_2$ (bottom panel) at …"

We thank the reviewer for spotting this. We put them the wrong way around. We have now changed the caption so that it matches the figure.

13. Page 15, line 351-352: "… with slopes (in ppm year$^{-1}$ and per meg per year$^{-1}$ for $CO_2$ and $O_2$, respectively) …" "…each TT, for $CO_2$ (top panel) and $O_2$ (bottom panel) …"

We thank the reviewer for spotting this. We put them the wrong way around. We have now changed the caption so that it matches the figure.

14. Page 27, line 618-619: "Manning, 2001" is not listed in References.

We have added this reference to the reference list:

Manning, A. C.: Temporal variability of atmospheric oxygen from both continuous measurements and a flask sampling network: Tools for studying the global carbon cycle, Ph.D. thesis, University of California, https://cramlab.uea.ac.uk/Publications.php, 2001.

15. Page 33, line 725-726: It would be better to clarify what the ranges in the parentheses mean. Are they 95% confidence intervals?

Yes, this is correct. We have amended the sentence as follows:

On average, atmospheric $CO_2$ at WAO increased by 2.40 ppm yr$^{-1}$ (2.38 to 2.42; 95% confidence intervals), atmospheric $O_2$ decreased by 24.0 per meg yr$^{-1}$ (24.3 to 23.8) and APO decreased by 11.4 per meg yr$^{-1}$ (11.7 to 11.3).

16. Page 35, line 767-768: I think that the effect derived from seasonal and/or diurnal covariance between surface fluxes and atmospheric transport including PBL dynamics is termed as rectification effect. The seasonal cycle of PBL height itself isn't termed as the rectification effect.

We have removed the phrase "called the seasonal rectifier effect" from this sentence and have changed it from:

> There is also a seasonal cycle in the height of the atmospheric boundary layer, called the seasonal rectifier effect, that influences all three species (Stephens et al., 2000).

To:

> There is also a seasonal cycle in the height of the atmospheric boundary layer, that influences all three species (Stephens et al., 2000).

17. Page 46, line 1036-1037: "Stephens, B. B., …, 2000" has been already listed in line 1033-1035.

We thank the reviewer for noticing this error. We have removed the duplicate reference.

---

## Referee Report (RR1)

**Review's comments**

**Manuscript Number:** essd-2023-129

**Title:** 12 years of continuous atmospheric $O_2$, $CO_2$ and APO data from Weybourne Atmospheric Observatory in the United Kingdom

**Authors:** K. E. Adcock, P. A. Pickers, A. C. Manning, G. L. Forster, L. S. Fleming, T. Barningham, P. A. Wilson, E. A. Kozlova, M. Hewitt, A. J. Etchells, and A. J. Macdonald

**Minor comments:**

Page 32, line 683-685: The larger magnitude of the atmospheric $O_2$ decreasing rate than the atmospheric $CO_2$ increasing rate is partially derived from the fact that the $-O_2/CO_2$ exchange rate for the globally averaged fossil fuel combustion is 1.38.

Several papers in References, Dlugokencky et al., 2023, Eddebbar et al., 2017, Friedlingstein et al., 2022, Rödenbeck et al., 2008, are not cited in the text.

The journal name of Pickers and Manning (2015) in References (Page 41, line 915) is missing.

---

## Author Response (AR2)

**Review's comments**

**Manuscript Number:** essd-2023-129

**Title:** 12 years of continuous atmospheric O2, CO2 and APO data from Weybourne Atmospheric Observatory in the United Kingdom

**Authors:** K. E. Adcock, P. A. Pickers, A. C. Manning, G. L. Forster, L. S. Fleming, T. Barningham, P. A. Wilson, E. A. Kozlova, M. Hewitt, A. J. Etchells, and A. J. Macdonald

**Minor comments:**

1. Page 32, line 683-685: The larger magnitude of the atmospheric O2 decreasing rate than the atmospheric CO2 increasing rate is partially derived from the fact that the $-O_2/CO_2$ exchange rate for the globally averaged fossil fuel combustion is 1.38.

We have added this into the sentence.

Atmospheric $O_2$ is decreasing more quickly than $CO_2$ is increasing because the globally averaged $O_2:CO_2$ exchange ratio for fossil fuel combustion is about 1.38 mol mol$^{-1}$ (Keeling and Manning, 2014) and the $CO_2$ increase is buffered by an increasing land carbon sink and ocean carbon sink, whereas the $O_2$ decrease is only buffered by an increasing land oxygen source and a small ocean oxygen source from $O_2$ outgassing (Keeling et al., 1996a).

2. Several papers in References, Dlugokencky et al., 2023, Eddebbar et al., 2017, Friedlingstein et al., 2022, Rödenbeck et al., 2008, are not cited in the text.

Thank you for spotting this. They were previously cited in the text in an earlier version of the manuscript. We have now removed them from the reference list.

3. The journal name of Pickers and Manning (2015) in References (Page 41, line 915) is missing.

Thank you for spotting this. We have added in the journal name.

Pickers, P. A. and Manning, A. C.: Investigating bias in the application of curve fitting programs to atmospheric time series, Atmos. Meas. Tech., 8, 1469–1489, https://doi.org/10.5194/amt-8-1469-2015, 2015.